# Therapeutic Potential of Autophagy Modulation in Cholangiocarcinoma

**DOI:** 10.3390/cells9030614

**Published:** 2020-03-04

**Authors:** Hector Perez-Montoyo

**Affiliations:** Ability Pharmaceuticals, SL, Cerdanyola del Vallés, 08290 Barcelona, Spain; hector.perez@abilitypharma.com; Tel.: +34-935-824-411

**Keywords:** cholangiocarcinoma, autophagy inhibition, autophagy activation, chemoresistance

## Abstract

Autophagy is a multistep catabolic process through which misfolded, aggregated or mutated proteins and damaged organelles are internalized in membrane vesicles called autophagosomes and ultimately fused to lysosomes for degradation of sequestered components. The multistep nature of the process offers multiple regulation points prone to be deregulated and cause different human diseases but also offers multiple targetable points for designing therapeutic strategies. Cancer cells have evolved to use autophagy as an adaptive mechanism to survive under extremely stressful conditions within the tumor microenvironment, but also to increase invasiveness and resistance to anticancer drugs such as chemotherapy. This review collects clinical evidence of autophagy deregulation during cholangiocarcinogenesis together with preclinical reports evaluating compounds that modulate autophagy to induce cholangiocarcinoma (CCA) cell death. Altogether, experimental data suggest an impairment of autophagy during initial steps of CCA development and increased expression of autophagy markers on established tumors and in invasive phenotypes. Preclinical efficacy of autophagy modulators promoting CCA cell death, reducing invasiveness capacity and resensitizing CCA cells to chemotherapy open novel therapeutic avenues to design more specific and efficient strategies to treat this aggressive cancer.

## 1. Introduction

Cholangiocarcinoma (CCA) is a very aggressive epithelial cell malignancy arising from varying locations within the biliary tree, a complex network of ducts that deliver bile to the gallbladder and to the intestine [1]. CCA originates from cholangiocytes located at any portion of the biliary tree and represents the most common biliary duct malignancy and the second most frequent cancer of the liver after Hepatocellular Carcinoma (HCC), accounting for 10–20% of all primary liver cancers [2,3,4].

The classification of CCA has been a matter of debate during the past decades, and depending on different aspects of these tumors, several classifications have been proposed. Based on the anatomy of the biliary tract and the different origins of the tumor, CCA is classified into three different types: intrahepatic cholangiocarcinoma (iCCA), which originates from the biliary tree within the liver proximal to the second-order bile ducts; and extrahepatic cholangiocarcinoma (eCCA), which originates outside the liver parenchyma. eCCA is further subdivided into perihilar cholangiocarcinoma (pCCA), arising between the second-order bile ducts and the insertion of the cystic duct into the common bile duct; and distal cholangiocarcinoma (dCCA), arising between the insertion of the cystic duct and the ampulla of Vater [2,5,6]. Although this anatomical classification is the most widely used, other factors such as tumor growth pattern (mass-forming, periductal infiltrating or intraductal) and the cell of origin (cholangiocytes, peribiliary glands, hepatic progenitor cells or hepatocytes) offer alternative classification that may be more useful in specific clinical settings [7,8,9,10].

CCA is a very deadly cancer which at an early stage remains asymptomatic and is normally diagnosed at advanced stages and in the elderly, where therapeutic options are reduced and have limited efficacy, showing high chemoresistance and death rates [2,11,12]. The only curative treatment is radical surgical resection and liver transplantation, which are limited to curing locally restricted disease [13,14]. However, most newly diagnosed patients present with advanced or even metastatic stages of disease, and chemotherapy is the only treatment option. Among all chemotherapeutic regimes available, only the combination of gemcitabine and cisplatin exerts some growth-inhibiting effects at advanced stages of the disease [15,16].

Autophagy is a multistep self-degradative cellular process in which misfolded, aggregated or mutated proteins and damaged organelles such as mitochondria, endoplasmic reticulum (ER) or peroxisomes are sequestered in double-membrane vesicles, which fuse with lysosomes for further degradation [17,18]. This tightly regulated process is important for maintaining nutrient and energy homeostasis and eliminating intracellular pathogens. Giving the housekeeping function of autophagy, it is generally a survival mechanism, but due to the multistep condition of the process and the multiple control points, autophagy can be deregulated at multiple sites, leading to multiple human diseases, including cancer [19]. Autophagy has been shown to act as a tumor promoter as well as a tumor suppressor in cancer, depending on the cell context, and autophagy modulation has arisen as a promising therapeutic strategy to treat cancer [20,21,22,23,24,25]. Even though the molecular mechanisms of autophagy regulation of tumor biology are not fully understood, multiple reports are showing promising therapeutic potential in combination with other drugs, such as chemotherapy [26].

In CCA, several reports released during the last decades have shown how autophagy deregulation is associated with malignant cells compared with normal cholangiocytes in clinical samples, correlating with metastatic disease and poor prognosis [27,28,29,30,31,32,33], and how autophagy modulation shows anticancer efficacy in preclinical models.

This review collects clinical and preclinical scientific reports involving autophagy modulation in CCA, putting all puzzle pieces together to try to shed light on the current knowledge of this therapeutic strategy for treating this aggressive disease.

## 2. Autophagy in Cancer

### 2.1. Autophagy Molecular Process

Macroautophagy (referred hereafter as autophagy), is a highly conserved catabolic process for recycling elderly, toxic or damaged intracellular components, mediated by the formation of autophagosomes that ultimately fuse to lysosomes for degradation [17,18]. The regulation and the roles of autophagy have been linked to almost all biological cell processes in both health and disease [19]. There are other less-studied forms of autophagy, including microautophagy, where cytoplasm components are engulfed through a tubular membrane invagination that fuses to lysosomes, and chaperon-mediated autophagy, where selected soluble cytosolic proteins are targeted to lysosomes. Autophagy can also be subclassified as nonselective autophagy, where cytoplasm is degraded in a bulk manner, and a less well-described selective autophagy, where autophagy selectively targets organelles and proteins for self-degradation (Figure 1), leading to generation of terms such as mitophagy (mitochondria degradation), pexophagy (peroxisomes degradation), lipophagy (lipids degradation) or xenophagy (microbe degradation), among others [34,35]. The formation and turnover of autophagosomes involve a conserved family of autophagy-related (ATG) genes, which are activated and recruited to membranes to initiate autophagy [36]. The main features of different types of autophagy as summarized in Table 1.

The autophagy process can be divided into distinct stages: initiation, nucleation of the autophagosome, expansion and elongation of the autophagosome membrane, fusion with lysosomes and degradation of intravesicular cargo [34]. In the initiation step, the Unc-51-like kinase1 (ULK1) complex is activated, a complex that includes ULK1, ULK2, ATG13, family interacting protein 200KD (FIP200) and ATG101. This ULK1 complex then phosphorylates and activates PI3K-Beclin1 complex, a class III PI3K complex formed by VPS15 (serine/threonine-protein kinase), VPS34 (a class III phosphatidylinositol 3-kinase (PI3K), ATG14 and Beclin1, or alternatively Beclin1 with UV radiation resistance associated gene protein (UVRAG or p63) and activating molecule in BECN1-regulated autophagy protein 1 (Ambra1), depending on the subcellular localization of the complex [37]. Beclin1 (Bcl-2 homology (BH)-3 domain only protein) is initially complexed with and inhibited by antiapoptotic protein Bcl-2, and upon different stimuli, this complex is disrupted and Beclin1 is released to initiate autophagy. ULK1 phosphorylates Beclin1, which acts as an overall scaffold for the PI3K complex facilitating localization of autophagic proteins into the phagophore [37].

This initial activation is coordinated by different inputs from the mechanistic target of rapamycin complex 1 (mTORC1) and AMP-activated protein kinase (AMPK). Under physiological nonstressed conditions, mTORC1 phosphorylates ULK1/2, keeping ULK complex inactive. When nutrient, energy, growth factors or other stress conditions affect the cells, mTORC1 is suppressed, and therefore ULK1 complex is dephosphorylated and activated. Activated ULK complex translocates to phagophore and induces vesicle nucleation by activating PI3K-Beclin1 complex [37]. These events lead to autophagosome formation following the extension and closure of the mature autophagosome. Two ubiquitin-like conjugation systems are main regulators for maturation, elongation and closure of the autophagosome membrane. On one side, ATG7 and ATG10 conjugate ATG5 to ATG12. ATG5-ATG12 forms a complex with ATG16L1. The ATG5-ATG12-ATG16L1 large multimeric (E3-like) complex gets anchored on the emerging autophagosomal membranes and recruits members of the microtubule-associated protein 1 light chain 3 (LC3) and GABARAP families to the autophagosome. On the other side, ATG7 and ATG3 conjugate the soluble form of LC3 (LC3-1) to phosphatidylethanolamine (PE), forming the lipidated form of LC3-I (LC3-II) on the surface of the emerging autophagosome guided by the ATG5-ATG12-ATG16L1 complex, which locates LC3-II on membrane to identify it as autophagic membrane and recruit more autophagic cargo through specific receptors [38]. LC3-II is often used in research as a marker for autophagy progression, since it localizes to both the inner and outer membranes of phagophores and autophagosomes and migrates faster than LC3-I on gel electrophoresis, allowing one to evaluate the ratio of lipidated LC3 to reflect the number of autophagosomes formed. The adaptor protein sequestosome 1 (p62) targets specific substrates to autophagosomes and is degraded along with other cargo proteins; therefore, it is normally used as a measure of autophagic flux [39].

At this point in the process, autophagosome is formed and is ready to internalize autophagic cargo and transport them on microtubules to the perinuclear region where lysosomes are present. Upon maturation, autophagosomes go into the last step in this catabolic process, the fusion of autophagosomes with lysosomes to form the autophagolysosome, a process that is regulated by three sets of protein families: the Rab GTPasas (Rab7 in autophagy), homotypic fusion and protein sorting–tethering complex (HOPS) and the soluble N-ethylmaleimide-sensitive factor attachment protein receptor (SNARE) proteins [40]. HOPS is a conserved protein complex consisting of vacuolar protein sorting 11 (Vps11), Vps16, Vps18, Vps33, Vps39 and Vps41 and mediates autophagosome–lysosome fusion through interaction with SNARE syntaxin 17 [41]. In this final step, UVRAG, which plays an important role facilitating Vps34 activation during initial steps of autophagy, shows a relevant role regulating autophagosome maturation in a Beclin1-independent manner. UVRAG recruits class C vacuolar protein sorting (C-Vps) complex to autophagosomes, where UVRAG-C-Vps interaction stimulates Rab7-GTPase activity that results in autophagosome fusion to lysosomes [42]. Lastly, in the degradation phase, autophagic cargo is degraded under the low pH of autophagolysosome that activates specific lysosomal hydrolases, recycling degraded material to be used to fuel growth of the cell (Figure 1).

Although autophagy was initially defined as a prosurvival cellular mechanism due to its role in maintaining homeostasis under stressful conditions, several reports have revealed its dual role in cancer [21] and the therapeutic potential of its modulation [20,21,22,23,24,25].

### 2.2. Autophagy as a Tumor Suppressor

Deficiencies in autophagy lead to the accumulation of impaired macromolecules and organelles that disrupt cell homeostasis and cause DNA damage and chromatin instability, key factors in the accumulation of oncogenic mutations. During the initial stages of malignant transformation, autophagy exerts a cytoprotective role, mainly acting as a tumor suppressor, lessening the effects of the metabolic stress and genome instability that cause tumor initiation [43,44]. Mostly, the inhibition of autophagy in cancer cells lies in the over-activation of the PI3K-Akt-mTORC1 pathway, which induces survival and proliferation [45]. Accordingly, several tumor suppressor genes such as PTEN [46], LKB1, AMPK [47] and TSC [48] are promoters of autophagy. Some of the most important evidence demonstrating the role of autophagy as a tumor suppressor comes from studies performed with Beclin1 [49]. Mice with genetic deletion of Beclin1 show a higher incidence of lymphoma, lung cancer and liver cancer [28]. In addition, monoallelic deletions of Beclin1 gene have been described in 40–75% of human cancers of the breast, ovary and prostate [50]. Consonant with these results, silencing of ATG5 results in the accumulation of p62 protein aggregates, defective mitochondria and poorly folded proteins, events that induce ROS (reactive oxygen species) production. An increase in ROS favors the appearance of potentially oncogenic mutations, and autophagy prevents malignant transformation by clearing accumulated p62 and limiting chromosome instability [43,51,52].

### 2.3. Autophagy as a Tumor Promoter

Activation of autophagy in established growing tumor cells is a common event among different types of cancers due to the extreme environmental conditions typical of the progressive tumor environment, such as lack of oxygen [53], limited nutrients [54] and increasing energy demand by sustained high metabolic rate [55]. Under these circumstances, autophagy appears as an adaptive cellular response that allows tumor cells to survive under severe conditions. RAS (Kirsten rat sarcoma viral oncogene homolog)-mutated cells are highly dependent on autophagy and are defined as “addicted to autophagy”. Oncogenic mutations in RAS are found in about 30% of human cancers and are tumors with high proliferative and metastatic potential [56,57]. Several studies have described that these cells depend on autophagy activation to maintain oxidative metabolism and glycolysis underpinning growth, survival, invasion and metastasis [58,59]. Autophagy is also presented as a protective strategy for tumor cells to evade the effect of various therapies and promote chemoresistance and tumor survival [60,61,62,63]. Drugs such as tamoxifen [64], temozolomide [65], resveratrol [66] or arsenic trioxide induce protective autophagy in cancer cells of the breast, prostate, colon and malignant glioma [67]. Radiotherapy has also shown induction of protective autophagy [68]. In many cases, the activation of autophagy has been linked to the development of resistance to these treatments. In this line, it has been described that the combination of autophagy inhibitors with chemotherapy, radiotherapy, tyrosine kinase receptor inhibitors or hormone therapy sensitizes cells to these treatments [67,68].

## 3. Cholangiocarcinoma Genetic and Epigenetic Alterations and Autophagy

CCA is a very heterogeneous group of malignancies highly influenced by different risk factors and genetic and epigenetic alterations [69]. Surgery, chemotherapy and locoregional therapy are the only approved therapies for CCA, although less than one-third of the patients have been classified as having a resectable tumor at the time of diagnosis. Tumor resection is usually followed by adjuvant chemotherapy using gemcitabine, cisplatin or 5-FU (5-fluorouracil), which nevertheless does not prevent the high rates of relapse and resistance. For patients presenting with unresectable or metastatic CCA, systemic chemotherapy remains the mainstay palliative treatment modality, and only gemcitabine plus cisplatin combination has offered limited advantages [15,16], usually followed by a fluoropyrimidine-based regimen when gemcitabine-based treatment fails [69]. The identification of genetic and epigenetic alterations and the increased knowledge about the molecular pathophysiological mechanisms governing cholangiocarcinogenesis and tumor recurrence, resistance and metastasis have allowed the development of more specific therapies, although clinical results evaluating specific molecular agents demonstrate no or only very modest survival benefits of the agents tested [4,5,70].

Whole-genome analyses identified two distinct genomic classes of iCCA: an inflammatory class with predominant activation of inflammatory pathways, and a second proliferative class with predominant activation of oncogenes that correlate with worse patient outcome [71]. Next-generation sequencing analysis revealed that the majority of CCAs showed a driver gene mutation, although tumors from different sites (iCCA versus pCCA and dCCA) have different genetic profiles. For example, RAS appears frequently mutated in CCA, with a higher prevalence in dCCA [72]. Exom sequencing analysis identified a unique subtype of CCA without RAS mutation and/or FGFR2 fusion genes [73]. Epigenomic studies have revealed that epigenetic modification such as DNA hypermethylation, histone modifications and microRNAs deeply affects CCA development [74]. All these data support the complexity of this type of cancer and the low efficacy of current diagnostic methods and therapies, and deeper research into the mechanisms leading to CCA establishment and progression will help to support the development of novel treatments that could improve therapeutic outcome based on proper patient classification.

Chronic inflammation, partial bile flow obstruction (i.e., cholestasis) and bile duct injury are recognized to be major features for malignant transformation [75]. Upon chronic inflammation, both cholangiocytes and immune cells secrete pro-inflammatory cytokines such as IL-6, endotoxins or TNF-α. Sustained IL-6 production acts as a key player in hepatobiliary inflammation and cancer development, promoting mitogenic responses and cell survival [76]. Additionally, IL-6 can increase nitric oxide synthase (iNOS)-mediated nitric oxide production, resulting in DNA damage [77] and cyclic oxygenase (COX)-2-mediated prostaglandin secretion that results in cell growth, antiapoptosis and angiogenesis [78]. Autophagy plays a relevant role in inflammation, although understanding of this interconnection is still incomplete [79]. Many of the signaling pathways that control inflammation during tumorigenesis are also known regulators of autophagy. For example, in lung cancer cells exposed to arsenic, oncogenic transformation correlates with sustained upregulation of IL6 and reduced autophagy [80], and IL-6-dependent transformation requires inhibition of a Beclin1-Bcl2 complex, which is dependent on STAT3 signaling. Moreover, enhancement of autophagy via Beclin1 overexpression is sufficient to block IL-6 mediated transformation [80]. This correlation between IL-6-mediated carcinogenesis and autophagy may represent an interesting and promising approach to treat iCCA with an inflammatory component. Additionally, there are a large number of studies that relate different pro-inflammatory pathways with ER stress and autophagy [79,81,82].

To date, different genes have been related to cholangiocarcinogenesis. Activating KRAS mutations can be found in up to 40% of CCAs, with major prevalence in dCCA and associated with a worse prognosis [72]. In a small study on 54 clinical samples of iCCA, 7.4% of cases were KRAS mutated and associated with higher tumor stage and worse long-term overall survival, as well as a greater likelihood of lymph node involvement [83]. Moreover, in a murine model of iCCA development harboring KRAS mutation and p53 inactivation, two of the most common genetic alterations in CCA [72,84], KRAS mutation collaborates with p53 deletion to cause hepatic transformation and reduced survival [85]. This murine model recapitulates histopathologic features of human iCCA and shows high basal levels of autophagy associated with tumor growth. Inhibition of autophagy with chloroquine (CQ) inhibited the growth of these cells and accumulated LC3-II, indicative of an active autophagy directly involved in tumor progression [85]. This data correlates with human iCCA cell lines mutated in KRAS and with p53 deficiency, which show elevated autophagy compared with normal iCCA cells, and CQ also inhibited the growth of these cells [86], similar to the situation described for pancreatic and lung cancers [56,87,88,89,90,91]. No specific RAS inhibitors have been developed so far, and targeted therapies aiming to modulate KRAS downstream pathways such as MEK1/2 inhibitor selumetinib are in development for CCA, pointing to the potential combination with autophagy inhibitors to improve their therapeutic potential [4,92].

Alterations in c-MET, the overactivation of which leads to activation of MAPK, PI3K/Akt and STAT pathways, correlates with high grade, invasiveness and poor prognosis in CCA [93,94], and its inhibition promoted autophagy in lung cancer cells [95], further linking c-MET-mediated autophagy inhibition in carcinogenesis. The gain of function mutation in ERBB2 and EGFR genes correlates with malignancy in human cholangiocytes, cancer progression and poor survival [96,97], and treatment with tyrosine kinase inhibitors induced protective autophagy in different cancer types [98], suggesting that the combination with autophagy inhibitors could increase the efficacy of these compounds. Similarly, FGFR2 fusion genes that result in altered cell morphology and increased cell proliferation have been described in CCA [99]. It has been shown that FGFR alterations suppress autophagy, which could be associated with initial steps of carcinogenesis, and genetic or pharmacological FGFR inhibition in vitro induces protective autophagy in lung and breast cancer; therefore, inhibition of autophagy increases anticancer efficacy of FGFR inhibitors in these cells [100,101]. There are currently FGFR inhibitors in clinical development for CCA, opening the possibility of evaluating the combination of these inhibitors with autophagy modulators to increase efficacy. Loss of SMAD4 is also frequently observed in CCA in the distal common bile duct [102], and it has also been shown to render pancreatic cancer radioresistance through promotion of autophagy [103]; hence, a combination with autophagy inhibitors also could potentially apply to these mutated tumors. Adenomatous Polyposis Coli (APC) is an additional tumor suppressor commonly mutated in CCA and may be responsible for the early stages of carcinogenesis [104], stages where dysfunctional autophagy has also been detected in clinical samples [105] and in xenografts during tumor formation [106].

Additionally, it has been proposed that epigenetic changes such as histone modifications, DNA methylation and noncoding RNAs, which play a very relevant role in the pathophysiology of CCA [107], are also regulators of autophagy [108]. Overexpression of histone deacetylase 6 (HDAC6) was reported in CCA, promoting the shortening of the primary cilium and inducing hyperproliferation. HDAC6 inhibition restores ciliary expression and decreases tumor growth in CCA [109,110], a mechanism that has been shown to be mediated by autophagy inhibition in colorectal cancer, multiple myeloma and neuroblastoma [111]. Other HDACs, such as HDAC1, have been found overexpressed in CCA and correlate with malignant behavior and poor iCCA prognosis [112]. Histone methylations also control autophagic flux, and it has been proposed that histone methylation keeps the brakes on autophagy [113]. DNA-methylation-mediated silencing of tumor suppressor genes is often seen in CCA. Frequent mutations in both DNA methylation IDH1 and IDH2 have been reported in 10% of iCCA, which are associated with hypermethylation of CpG shore, resulting in an altered state in the cellular process of differentiation [114,115]. Several reports highlight the link between autophagy inhibition and histone methylation [108,113], suggesting autophagy inhibition as a target for treating IDH mutant gliomas [116]. A number of microRNAs (e.g., miR-141, miR-200b, miR-21, miR-29b among others) have been described to be either up- or downregulated in CCA cell lines, and their predicted targets were found to be associated with cell growth, apoptosis and response to chemotherapy in CCA cell lines [117,118]. MicroRNAs are also involved in regulating autophagy in cancer, and different autophagy-related proteins have been described as miRNAs targets, such as ULK2, Beclin1, LC3, ATG4 and ATG9 [119,120]. Moreover, miR-124 has been described to induce cytotoxic autophagy in CCA through the EZH2–STAT3 pathway in vitro and in vivo [29].

## 4. Autophagy Modulation in Cholangiocarcinoma

Although the pathologic role of autophagy in cholangiocarcinogenesis and the therapeutic potential of its modulation are still poorly understood, several reports have identified autophagy-related markers with prognostic significance, underlining the relevance of this process in CCA and offering novel therapeutic avenues.

Similar to pancreatic cancer, CCA follows a carcinogenic development in which a precursor lesion, a biliary intraepithelial neoplasia (BilIN), is developed. The study of the expression levels of LC3, Beclin1 and p62, along with p53 and KRAS status on clinical BilIN samples and compared with normal bile duct and peribiliary gland, revealed that autophagy deregulation may occur at an early stage of development of CCA [105]. Expression of LC3 and p62 was high in BilIN stages 1-2 compared with normal cholangiocytes, and LC3, Beclin1 and p62 were all higher in invasive carcinoma compared with nontumoral tissue. No significant correlation between KRAS and expression of autophagy markers in BilIN 1-2 stages was observed. Autophagy is a dynamic process, and accumulation of LC3-II and p62 in initial steps of cholangiocarcinogenesis could reflect a defect in the later processing of autophagosomes rather than increased rates of autophagy. This would correlate with the tumor suppressor role of autophagy in these initial steps, where its inhibition could permit carcinogenic transformation of cholangiocytes.

Epithelial to Mesenchymal Transition (EMT) is considered to be a major driver of cancer exacerbation, promoting tumor progression, metastasis and drug resistance [121,122]. The link between EMT and autophagy has been amply demonstrated, since main pathways regulating autophagy have a dramatic impact on EMT, such as PI3K/AKT/mTOR, Beclin1, p53 and JAK/STAT signaling pathways. Additionally, signaling pathways implicated in EMT are crucial in autophagy, including integrins, WNTs, NF-kB, and TGF-β signaling pathways [123]. In CCA, EMT leads to immunosuppression through SNAIL expression [124] and is critical for invasiveness and metastasis induced by TGF-β1/SNAIL activation [125]. Autophagy inhibition with CQ reduced invasive capacity under starvation and in TGF-B1-induced CCA cell invasion [126], further exposing EMT and autophagy relation in CCA and reinforcing the idea of a tumor promoter role of autophagy in established CCA tumors.

Beclin1 plays a relevant role linking autophagy, apoptosis and differentiation, and its inactivation and consequent deficiency in autophagy was correlated with malignant transformation, although existing data on the prognostic role of Beclin1 in human carcinomas is contradictory, appearing under- and overexpressed in distinct human cancers [49,127,128]. Several studies have shown the significance of Beclin1 in iCCA [27,28] and eCCA [28], revealing its potential prognostic value for CCA. Beclin1 was found markedly expressed in iCCA samples compared with normal bile duct epithelium [27], and among Beclin1-positive samples, those with low Beclin1 expression were significantly associated with lymph node metastasis, worse overall survival and less disease-free survival [27,28]. Moreover, in a lymph-node-negative CCA subgroup, Beclin1 was higher than in the lymph-node-positive subset, suggesting that Beclin1 inactivation and therefore impaired autophagy might promote malignant phenotypes. Interestingly, a stratified survival analysis in patients with Beclin1 low expression, iCCA patients showed a worse overall survival and progression-free survival than eCCA [28], which may indicate a higher implication of autophagy in iCCA subgroup of patients. Nevertheless, low Beclin1 levels show a correlation with poor prognosis in both subtypes [28]. This clinical data is in contradiction with other reports that indicate an exacerbated autophagy in CCA samples and its association with lower survival and tumor dissemination. Ambra1, a positive regulator of the Beclin1-dependent program of autophagy, positively correlated with SNAIL expression in CCA patients. SNAIL is a hallmark of EMT activation, which is in accordance with the in vitro increased invasive potential mediated by autophagy in TGF-β1/SNAIL-induced EMT [126]. These opposing results underscore the need to clearly define the type of studies that would help to discern whether the presence of autophagy-related markers are associated with impaired or increased autophagic flux, and additional expression studies of other markers such as LC3-II, p62, PI3Ks or ATGs could add significant value.

In another recent study, Chen and colleagues demonstrated for the first time that LC3B is an independent biomarker for overall survival and progression-free survival in iCCA patients, and that high LC3B staining significantly associates with poor tumor differentiation, tumor stage, early relapse and bad long term survival. Based on nomograms, they stratified iCCA patients and generated a therapeutic strategy after hepatectomy, demonstrating that nomograms based on autophagy markers can be considered as effective models to predict postoperative survival of iCCA patients [31]. In a very interesting study published in 2019, Atg7 was found to be a causative genetic risk factor for CCA development in a family with a high incidence of pCCA, identifying a germline mutation associated with CCA development [33]. This genetic variant resulted in the accumulation of p62, indicative of impaired autophagy in the tumors of carriers compared with noncarrier tumors, confirming autophagy pathway perturbation as a novel cancer driver mechanism in human tumorigenesis in correlation with the detection of impaired autophagy in BilIN lesions [105].

Another potential therapeutic target associated with autophagic flux in CCA is FOXO1. FOXO1 expression and transcriptional activity are involved in promoting cellular autophagy, and the interaction of acetylated FOXO1 with ATG7 regulates basal and starvation-induced autophagy in CCA cells [30]. Cytoplasmic accumulation of FOXO1 is associated with increased proliferation in cholangiocytes [129] and pharmacological inhibition of acetylated FOXO1, which results in autophagy inhibition, leads to apoptosis induction and reduced viability of CCA cells [30]. Epigenetic alterations are frequent in CCA, such as miR-124, which was found significantly downregulated in the tumor tissue of patients and in CCA cell lines, and its administration in vitro induced cytotoxic autophagy in CCA cells [29], supporting a protumoral role of epigenomic-mediated inhibition of autophagy.

## 5. Clinical Development of Autophagy Modulators in Cholangiocarcinoma

Multiple clinical trials are currently ongoing testing the efficacy of different anticancer drugs on CCA patients administered alone or in combination. A search for phase II and III trials was operated on clinicaltrial.gov (data of entry 2020-01-15) combining terms such as cholangiocarcinoma, autophagy, mTOR, AKT, PI3K, chloroquine and hydroxychloroquine and obtaining a limited set of studies. Two different trials are exploring the inhibition of autophagy in CCA using CQ (NCT02496741-completed; [130]) and hydroxychloroquine (HCQ) (NCT03377179-recruiting). The study involving CQ explores safety, recommended phase 2 dose and efficacy of metformin and CQ combinatory treatment in IDH1/2 mutated solid tumors, alteration found in around 20% of iCCA patients. This could seem contradictory, given the fact that metformin, an approved antidiabetic drug, is considered to act by inducing AMPK-mediated autophagy, although its mechanism of action is still far from being completely understood. The study using HCQ combines this autophagy inhibitor with ABC294640 (Opaganib), a first-in-class sphingosine kinase-2 (SK2)-selective inhibitor. ABC294640 was proven to induce protective autophagy in cancer [131], and this study relies on the HCQ-mediated potentiation of ABC294640 anticancer activity by inhibiting ABC294640-mediated protective autophagy in CCA.

When looking at mTOR inhibitors as autophagy inducers in CCA, preclinical evaluation of everolimus (RAD001) showed a reduction in cell proliferation with increased apoptosis and decreased invasion [132], although no reference to autophagy is clearly shown in spite of the association of PI3K/AKT/mTOR signaling pathway with CCA metastasis [133]. PI3K/AKT/mTOR inhibitors in clinical development for CCA include mTOR, PI3K and AKT inhibitors administered alone or in combination with chemotherapy. Among mTOR inhibitors, Everolimus is administered as monotherapy (NCT01525719—unknown and NCT00973713—unknown), in combination with gemcitabine and oxaliplatin (NCT02836847—recruiting) and with FOLFIRINOX (NCT03768375—recruiting), and sorafenib is administered alone (NCT00238212—completed), in combination with gemcitabine and cisplatin (NCT00919061—completed), with gemcitabine and oxaliplatin (NCT00955721—terminated and NCT02836847—recruiting), with erlotinib (EGFR inhibitor) (NCT01093222—completed) and with FOLFIRINOX (NCT03768375—recruiting). Two studies using MK-2206 AKT inhibitor were found administered as monotherapy (NCT01859182—terminated and NCT01425879—completed) and one with BKM120 PI3K inhibitor as monotherapy (NCT01501604—terminated).

Current clinical evaluation of autophagy modulators is still missing, probably due to the lack of knowledge about the mechanism that could lead to a synergistic effect. Only CQ and HCQ are been clinically evaluated, and results from these trials, specially HCQ combination with ABC294640, will be of great interest to obtain initial conclusions of the therapeutic potential of inhibiting autophagy to increase the efficacy of protective-autophagy-inducing anticancer drugs. Nevertheless, further research is needed to try to get accurate patient selection in order to increase efficacy.

## 6. Autophagy Modulators in Cholangiocarcinoma

### 6.1. Autophagy Inhibitors

Due to the dual role of autophagy in cancer cells, its modulation either by activation or by inhibition has emerged as a promising therapeutic strategy for cancer treatment. Within the strategies to inhibit autophagy in cancer, several compounds target different steps of the autophagic process, such as ULK inhibitors, pan PI3K inhibitors, VPS34 (PI3KC3) complex inhibitors, ATG inhibitors, autophagosome formation inhibition and lysosome inhibitors [23,24,25]. In CCA, several publications show the anticancer efficacy of autophagy inhibitors using different approaches. Three studies reported CQ efficacy on CCA cells, an antimalaria drug that inhibits the last step of autophagy, blocking autophagosome fusion with lysosomes [134,135,136]. GNS561 is a lysosomotropic small molecule that also blocks fusion of autophagosomes to lysosomes by altering the acidic pH of lysosomes [137]. Several natural compounds are under evaluation in CCA preclinical models. Salinomycin (a naturally occurring polyether antibiotic [138]), capsaicin (a major pungent component of chili pepper [139]), oblongifolin C (a natural small molecule extracted from *Garcinia yunnanensis* Hu [140]) and resveratrol (a natural phenol, phytoalexin, produced by plants against infections [30]) have shown anticancer efficacy on CCA models by different mechanisms: inhibiting autophagosome fusion to lysosomes, promoting mTOR activation and blocking ATG7 activation, respectively. Two class III PI3K inhibitors that block initiation of autophagy (3-MA and wortmannin [106]) and Mdivi1 (selective Drp-1 inhibitor [141]), which interferes with mitochondrial activity, have also shown efficacy on CCA.

Hou and colleagues published in 2011 that CCA clinical samples showed higher autophagic vacuole content and increased expression of Beclin1 and Atg5 compared with normal cholangiocytes. Interestingly, they found induction of autophagy in human CCA cell lines under starvation and during tumor formation in xenograft models, suggesting a potential role of autophagy in CCA tumorigenesis and the therapeutic potential of its inhibition. In correlation with this, genetic beclin1 depletion or pharmacological inhibition of autophagy by inhibiting PI3K-Beclin1 complex with 3-MA (3 methyl adenine) and wortmannin hampered proliferation and increased apoptosis during nutrient starvation, sensitizing iCCA cells to chemotherapeutic-agent-induced cell death in vitro and in vivo accompanied by a decrease in ATG5 and Beclin1 mRNA levels [106].

Among natural compounds that inhibit autophagy in CCA, capsaicin is the only one that induces autophagy inhibition through mTOR activation. Capsaicin interferes with NF-kB and AP-1 signaling, resulting in negative regulation of cell survival, adhesion, inflammation, differentiation and growth, and although it showed induction of autophagy in melanoma [142], it inhibits autophagy in CCA by activating PI3K/AKT/mTOR pathway, increasing sensitivity of CCA cells to 5-FU [139]. Zang and colleagues reported in 2016 the use of oblongifolin C as an autophagy inhibitor that blocks the autophagosome fusion to lysosomes and promotes mitochondrial dysfunction (MyD), leading to apoptosis induction [140]. Moreover, pharmacological enhancement of autophagy impaired oblongifolin C effects and treatment with 3-MA potentiated its anticancer effects, reinforcing the implication of the inhibition, although much research is needed to fully understand its precise mechanism of action. Salinomycin is another natural compound whose mechanism of action is still unclear, but it has been reported to have anticancer activity in CCA by inhibiting autophagy. This antibiotic interferes with Wnt signaling, inhibiting autophagic flux, which leads to the accumulation of dysfunctional mitochondria and increased generation of ROS, suggesting it can affect the fusion of autophagosomes with lysosomes in a similar way to CQ [138]. Moreover, salinomycin inhibited KRAS and p53 mutated CCA tumor grothw in vivo, in correlation with the potential use of this strategy to treat KRAS-driven tumors. Resveratrol, which has been shown to induce autophagy-mediated cell death in leukemia and gastric cancer cells [143,144], showed autophagy inhibition in CCA by promoting deacetylation of FOXO1, impairing FOXO1 binding to Atg7 and blocking autophagy initiation in CCA cells, finally leading to apoptosis [30]. Moreover, cytoplasmic accumulation of FOXO1 is associated with increased proliferation in cholangiocytes [129], further validating the role of FOXO1 in the initiation step of autophagy. Two additional reports published in 2018 used GNS561 and MdIvI-1 as therapeutic autophagy inhibitors in CCA. GNS561 promotes lysosomal dysregulation through lysosome permeabilization and release of hydrolytic enzymes to the cytosol, leading to the impairment of autophagosome fusion to lysosome and induction of apoptosis in vivo in iCCA xenografts [137]. Mdivi-1 is thought to act inhibiting enlongation of autophagosomes impeding mitochondrial dynamics, leading to autophagy inhibition that potentiates cisplatin-induced apoptosis in CCA [141].

CQ is the autophagy inhibitor most widely used in cancer, and currently the only autophagy modulator (except from PI3K/AKT/mTOR inhibitors) under clinical evaluation for CCA. In CCA models, CQ attenuates invasive activity of CCA cells under starvation, reducing TGF-β1-induced CCA cell invasion [134] and sensitizing resistant CCA cells to cisplatin [135]. CQ acts altering the acidic environment of lysosomes, blocking blocking their binding with autophagosomes, which results in accumulation of a large number of degraded proteins in the cytoplasm and the induction of ER stress. This sustained ER stress activates CHOP, which finally induces the activation of multiple death-signaling pathways in CCA, including caspase 3 and 8, cleaved PARP and Bcl-2 family proteins Bax and Bak [136].

Activation of autophagy as a resistance mechanism in response to chemotherapy has been widely described for many different types of cancers, including CCA [60,61,62,63]. A wide variety of anticancer compounds induce autophagy in CCA, making it necessary to discern whether it is a protective autophagy promoted by cancer cells as an adaptive mechanism, therefore inhibition of autophagy leads to a potentiation of their cytotoxic effects, or if on the contrary, mediates drug mechanism of cancer cell death induction. Several compounds that show anticancer efficacy on CCA cells such as norcantharidin [145], compound C [146], vorinostat [147] or cisplatin [106,141] induce the activation of protective autophagy in CCA cells, and pharmacological inhibition of autophagic process enhances these drugs anti-cancer capacity, accelerating apoptosis and sensitizing cell to chemotherapy. The combination of these drugs with autophagy inhibitors offers an attractive therapeutic strategy. Following this rationale, currently recruiting clinical trial combining HCQ with SK2 selective inhibitor ABC294640 in CCA patients attacks cancer cells inhibiting ABC294640-induced protective autophagy, with the aim to increase efficacy in these patients. This is a very promising strategy to apply to other combination that has already shown preclinical efficacy.

### 6.2. Autophagy Activators

There are several strategies currently under evaluation to induce autophagy-mediated cell death in cancer, including mTOR inhibitors; BH3 (Bcl-2 homology 3) mimetics, which promote the liberation of Beclin-1 from Bcl2 and Bcl-XL inhibition [148]; cannabinoids, which induce an exacerbated ER stress on cancer cells ultimately leading to CHOP-mediated apoptotic cell death [149]; HDAC inhibitors, which act through the epigenetic modulation of autophagy [22,150] and natural compounds extracted from plants, herbs or insects [22,150].

Four natural compounds have been recently described to induce CCA cell death, implicating the activation of autophagy as a mediator of their cytotoxic effects:piperlongumine (small molecule extracted from Piper longum plant [151]), pterostilbene (an active constituent of blueberries [152]), pristimerin (a triperpenoid isolated from Maytenus heterophylla [153]) and dihydroartemisinin (an active compound found in Artemisia annua [154]). Although it has been proven that autophagy induction is necessary for their mechanism of action, the specific molecular mechanisms governing their autophagy modulation abilities are not fully understood yet. Piperlongumine induced apoptosis [155] and autophagy [151] in CCA cells through the production of ROS, induction of ER stress and activation of JNK-ERK signaling pathway [151]. Similar to piperlongumine, dihydroartemisinin is an antimalaria drug that induces ROS-mediated ER stress through DAPK activation, promoting the disruption of Beclin1-Bcl-2 complex and inducing autophagy-mediated CCA cell death, therefore activating initiation of autophagy. Importantly, its cytotoxic effects were cancer-cell-specific, since only slight toxicity was observed on immortalized cholangiocytes. Beclin1 activation is crucial for dihydroartemisinin action since its genetic depletion or its pharmacologically-mediated degradation inhibits autophagy activation and partially protects CCA cells from dihydroartemisinin treatment [154]. Another drug that promotes Beclin1 activation is pristimerin, which inhibited CCA cell growth in vitro and in vivo, decreasing apoptosis-related proteins Bcl-2, Bcl-XL and procaspase-3, similar to the effect of BH3 mimetics, suggesting pristimerin promotes Beclin1 activation and initiation of autophagy. Interestingly, this compound showed higher efficacy on eCCA cell line QBC939 versus iCCA cell line REB, making it attractive to further investigate what mediates such selectivity [153]. Pterostilbene, a natural demethylated analog of resveratrol, induced inhibition of proliferation and clonogenicity of CCA cells in vitro and in vivo mediated by cytoplasmic vacuolation in an apoptosis-independent manner. Pterostilbene induced increased expression of p53, ATG5, Beclin1 and LC3 but decreased levels of p62, indicative of an active autophagy, suggesting it could act at the initiation steps promoting Beclin1 activation or autophagosome nucleation [142].

During recent years, four additional reports have been published using autophagy inducers in preclinical models of CCA. Decitabine (a cytosine analog, DNA demethylating agent [156]) and miR-124 (associated with STAT3 signaling) [29] induce an epigenomic induction of autophagy, while phenformin (diabetes therapeutic biguanide compound [157]) and ABTL0812 (hydroxylated variant of linoleic acid) [158] induce autophagy-mediated CCA cell death by activating LKB1-AMPK pathway and by inducing ER stress activation and AKT/mTOR pathway inhibition, respectively. Decitabine can potentially modulate the response of cancer cells to chemotherapy and radiotherapy [159] and induced apoptosis and autophagy-dependent caspase-independent CCA cell death in vitro, reducing tumor growth in vivo [156]. While pristimerin showed more efficacy on iCCA versus eCCA cells, decitabine showed different efficacy among two different eCCA cell lines, suggesting the induction of autophagy with this compound may be related to cell-specific characteristics rather than to the morphologic origin of CCA [156]. Another epigenetic factor, miR-124, induces a tumor-suppressive effect in CCA by inducing autophagic flux, leading to autophagy-related cell death in a mechanism involving EZH2–STAT3-signaling axis. Silencing of Beclin1 or ATG5 abrogated miR-124 anticancer effects and its overexpression in xenograft models resulted in autophagy-mediated suppression of tumorigenicity through STAT3 activation, Bcl-2 downregulation and Beclin1 expression, which indicates that it acts at the initiation of autophagy. Moreover, miR-124 was downregulated in human CCA samples compared with nontumor tissue [29]. Another approach to induce autophagy in CCA cells has been through the activation of the LKB1–AMPK pathway, leading to mTOR inhibition by phenformin. Hu and colleagues showed that phenformin inhibits complex 1 of mitochondria, increasing intracellular AMP and inducing the activation of LKB1–AMPK axis, leading to mTOR inhibition. As a consequence, apoptosis and autophagy are increased, along with an increase in ATG7, ATG5 and Beclin1 levels, therefore acting on mTOR-mediated ULK1 complex activation during initiation of autophagy. The last published report precisely determined the mechanism of action of ABTL0812, which induces cytotoxic autophagy on CCA cells by inducing robust and sustained ER stress [158,160], along with TRIB3-mediated Akt/mTOR axis inhibition [161]. Similar to dihydroartemisinin, at ABTL0812 concentrations that result in lethality for CCA cells in vitro, immortalized cholangiocytes remain alive, suggesting that this type of anti-cancer treatments may offer a safe approach.

Promotion of autophagy in response to cell stress conditions such as lack of growth factors or hypoxia activates autophagy via mTORC1 inhibition [162]. Additionally, other types of cell stresses promote autophagy through the UPR (Unfolded Protein Response) and mediated by PERK, IRE1α or CAMKK2 protein [163]. PERK activation directly activates the ATG12–ATG5–ATG16L complex, which induces PERK–ATF4–CHOP pathway activation and TRIB3 (Tribbles homolog 3) expression, a pseudokinase that acts as an endogenous negative regulator of the AKT/mTOR axis [158,164,165]. IRE1α promotes Beclin1 liberation from Bcl2 and PI3K–Beclin1 complex activation [163]. This is the case for some drugs such as tetrahydrocannabinol (THC), which exerts its antitumoral action by inducing ER-stress-mediated apoptotic cell death [149,165,166] and has shown anticancer efficacy in CCA [167]. ABTL0812 is the autophagy inductor currently being evaluated in CCA models with the most complete description of its mechanism of action, and it already showed preliminary clinical efficacy on a CCA patient derived from a phase I trial in patients with solid tumors [160,168]. In xenograft models, ABTL0812 potentiated gemcitabine plus cisplatin anticancer efficacy by upregulating TRIB3 and CHOP levels, two markers that have been validated for the first time as surrogate pharmacodynamic biomarkers in endometrial and lung cancer patients [158,160,169,170]. This novel strategy to induce ER-stress-mediated cytotoxic autophagy relies on the fact that cancer cells have evolved to use the UPR to survive the ER stress induced by the hostile conditions of tumor microenvironment (hypoxia, low glucose, intracellular acidification, etc.), exhibiting higher ER stress basal levels [171]). The induction of ER stress in cancer cells is a common mechanism of natural compound activators of autophagy and can result in an overpass of the cytoprotective effect of the UPR, leading to activation of the pro-apoptotic arm (CHOP) and to cell death. On the contrary, nontumoral cells show negligible levels of ER stress and therefore possess a broader margin to resist stress-induced cytotoxicity [172], correlating with lower cytotoxicity on immortalized cholangiocytes observed for ABTL0812 and dihydroartemisinin.

A summary of autophagy modulators in preclinical models of CCA is described in Table 2, and a graphic showing their mechanism of action within the autophagic process is illustrated in Figure 2.

## 7. Discussion and Future Perspectives

Autophagy is a tightly orchestrated multistep catabolic process generally considered a prosurvival mechanism, which allows cells to recover homeostasis under stressful conditions by controlling energy and nutrient balance [17,18]. The presence of multiple checkpoints within the autophagic process increases the possibilities of disturbing autophagy and developing different human diseases including cancer, although it also offers multiple target points for therapeutic approaches [19]. The precise molecular mechanisms linking autophagy and cancer cell fate are still to be determined, although numerous reports addressed to uncover these molecular mechanisms have been released during last decades. Autophagy may act as tumor suppressor at the early stages of cancer development, impeding the appearance of oncogenic mutations through the clearance of impaired macromolecules and organelles that cause DNA damage and chromatin instability [43,44]. When the autophagic process is impaired, the accumulation of p62 aggregates, defective mitochondria, poorly folded proteins and increased intracellular ROS promote malignant transformation [43,51,52].

In CCA, several pieces of evidence strongly suggest a deregulated autophagy at the initial steps of cholangiocarcinogenesis, where defective autophagy would allow oncogenic transformation. Supporting this theory, Greer et al. showed a genetic risk of CCA linked to ATG7 deficiency and therefore autophagy impairment, mediated by a lack of lipidation activity and p62 accumulation compared with wild-type ATG7 carriers [33]. Moreover, precursor BilIN lesions showed higher levels of LC3-II and p62 compared with normal biliary ducts [105], indicative of uncomplete autophagic process, reinforcing the theory of autophagy inhibition as a contributor to carcinogenic transformation. Several genetic alterations commonly observed in CCA have also been linked to autophagy inhibition in other types of cancers, such as c-Met [95], FGFR gain of function [100,101] or HDAC6 overexpression [111]. These genetic alterations could mediate cholangiocyte oncogenic transformation through the inhibition of autophagy, cooperating with their proliferative and prosurvival-derived effects. It has been demonstrated that continuous IL-6 secretion mediated by STAT3 inhibits autophagy, contributing to arsenic carcinogenesis in lung cells during carcinogenesis [80], strengthening the idea of impaired autophagy during CCA establishment that could also take place in the inflammatory subtype of CCA.

Autophagy can also act promoting tumor growth on established tumors serving as an adaptive and pro-survival mechanism against the extreme tumor microenvironment conditions such as lack of oxygen, limited nutrients and high metabolic rate [53,54,55]. Thongchot and colleagues found a positive correlation between HIF-1α (hypoxia-inducible factor 1-α) with BNIP3 (pro-apoptotic member of Bcl2 family) and PI3KC3 (component of Beclin1-PI3K complex), which associated with poor prognosis and lymph node metastasis in CCA samples [173], reflecting an hypoxic stress that activates autophagy as prosurvival and invasive mechanism. Similarly, RAS-mutated cells have been defined as addicted to autophagy by maintaining oxidative metabolism and glycolysis, underpinning growth, survival, invasion and metastasis [58,59]. RAS appears frequently mutated in CCA [72], suggesting that these cells could also have high dependence on autophagy for survival. Supporting this idea, a transgenic murine model of iCCA carrying KRAS and p53 genetic alterations, showed actively engaged autophagy [85], as well as iCCA cells in vitro [91]. Treatment of primary cells derived from intrahepatic murine tumors with CQ led to LC3-II accumulation and induced cancer cell death, revealing an active autophagy in these tumors. Interestingly, when autophagy is impaired in these cells by ATG7 deletion, mice died from inflammation rather than from tumor-derived effects such as lung or liver metastatic, further reinforcing the idea of autophagy activation as a mediator of survival and growth in CCA [85,91].

The use of autophagy inhibitors such as HCQ or CQ arises as a very promising strategy to treat different cancers, especially those with autophagy dependence for growing and dissemination. In KRAS-driven cancers, autophagy-dependent production of secreted factors facilitates invasion [59], where EMT has a prominent role. EMT induced by TGF-β in CCA cells was shown to mediate a higher invasive capacity [125], and inhibition of autophagy impaired invasiveness in vitro mediated by EMT induction, which highlights the importance of autophagy for increasing CCA metastatic potential. Moreover, this in vitro data correlates with higher expression of autophagy-related markers in CCA patients with lymph node metastasis such as Ambra1 [126], which also correlates with SNAIL expression, a master regulator of EMT. Another advantage of inhibiting autophagy relies on the blockage of the protective mechanism mediated by autophagy activation induced by different drugs. A wide variety of anticancer compounds induce protective autophagy in CCA [145,146,147] including chemotherapy [106,141] and the inhibition of autophagy accelerated apoptosis and chemosensitized CCA cells. This opens up different possibilities to design combinatory treatments that could block this protective autophagy and enhance the therapeutic effects of different drugs in addition to diminishing tumor dissemination. This is the rationale behind a clinical trial currently ongoing for CCA patients, in which HCQ is administered in combination with a selective SK inhibitor (ABC294640) previously shown to induce protective autophagy in cancer [131]. Inhibiting autophagy would block the activation of autophagy as a mechanism of resistance and could potentially decrease CCA metastatic potential; therefore, clinical results of this study would be of great help for further design of novel therapeutic strategies involving autophagy inhibitors in CCA.

Beclin1 has been defined as a tumor suppressor and is a critical factor in autophagy initiation, directly interacting with prosurvival and prodeath factors, thus being involved in cell fate decision making [44,49,174]. In CCA, different reports analyzing the potential role of Beclin1 as a prognostic marker have been released, although showing some contradictory results. Beclin1 was found overexpressed in CCA samples compared with normal biliary duct cells, and within Beclin1-positive CCA samples, low Beclin1 was associated with poor prognosis and lymph node metastasis [27,28]. Interestingly, low Beclin1 expression was associated with poor prognosis and less overall survival in both iCCA and eCCA patients, although iCCA had an inferior overall survival compared with eCCA patients. In opposition to this data, Ambra1, a positive regulator of Beclin1, showed higher expression in CCA patients with lymph node metastasis and poor survival [126].

Similar to CCA, Beclin1 expression in different cancers is differently associated with prognosis, metastasis and survival [49,127,128]. In ovarian carcinomas, decreased expression of Beclin1 was correlated with histological grade, advanced clinical stage and shortened patient survival and inversely correlated with Bcl-xL expression, showing that the low Beclin1/high Bcl xL group had the lowest survival rate [175]. In breast carcinomas, low expression of Beclin1 may contribute to the development and progression of breast cancer [49]. Conversely, high beclin1 expression was found predictive of poor prognosis in nasopharyngeal carcinoma [176], and Beclin1 and LC3 high expression correlated with tumor stage, metastasis and survival in pancreatic [177] and colorectal [178] cancers. Recent studies addressing the potential of Beclin1 expression as a prognostic factor in different cancers have emphasized the necessity to combine Beclin1 expression with other autophagy-related proteins such as HIF-1α, Bcl2 family proteins Bcl-xL and BNIP3, PI3KC3 or ATGs to increase its clinical value.

In recent studies, the low Beclin1/high Bcl-xL population, but not the low-Beclin1/low-Bcl-xL population of HCC patients, was associated with the most aggressive disease and tumor differentiation [179], and similar results were observed between Beclin1 and apoptotic markers Bcl-2 and Bax [180] and between Beclin1 and HIF-1α [181]. In a histopathological retrospective study on iCCA clinical samples, ARID1A, CA9 and IDH1 were found highly expressed in iCCA tumor tissues, but only high Beclin-1/high ARID1A populations were strongly associated with poor prognosis, lower survival rate and a worse recurrence rate than patients with low Beclin-1/low ARID1A expression [182]. This recently published study seems to be in contradiction with previously published reports where low Beclin1 was associated with poor prognosis [27,28]. All these data underline a need for clearly defined specific marker combinations that could predict CCA prognosis, metastasis and survival, and that could potentially serve to stratify patients for specific combinatory treatments involving autophagy modulators. For example, expression analysis of p62 and LC3-II protein levels could significantly help to identify whether autophagy is engaged or impaired. Higher Beclin1 levels could indicate increased autophagic activity, but if it is accompanied with p62 accumulation it would indicate autophagy impairment, probably due to defects in last steps in autophagosome degradation. A good example of the usefulness of the detection of multiple markers in CCA was the positive correlation found of HIF-1α with BNIP3 and PIK3CA, indicative of high autophagic activity and related to poor prognosis [173], therefore positioning this population of patients as potential targets for autophagy inhibition therapeutics.

The induction of autophagy as a therapeutic approach to treat CCA is also showing promising results. The induction of ER-stress-mediated cytotoxic autophagy by increasing intracellular dihydroceramides (Dh-Cer) content has been proposed as a safe and efficient way to induce autophagy-mediated apoptosis in cancer cells. Resveratrol [144], which in CCA acts inhibiting autophagy, and THC [183] induce an increase in Dh-Cer in cancer cells by inhibiting dihydroceramide desaturase (Des-1), which leads to ER-stress-mediated autophagy promotion. Similarly, ABTL0812 induces impairment of Des-1 activity, resulting in the accumulation of Dh-Cer and activation of UPR response, which, in combination with TRIB3-mediated AKT/mTOR axis inhibition, triggers cytotoxic autophagy in CCA cells [158,160]. Interestingly, Des1 expression was found to be upregulated in CCA cell lines compared with their nontumor counterparts NHC3 cells [158], correlating with previous reports [184] where Des1 was found overexpressed in CCA tissue compared with normal biliary tract tissue. Cancer cells have evolved to use the UPR to survive the ER stress induced by the hostile conditions of the tumor microenvironment (hypoxia, low glucose, intracellular acidification, etc.); therefore, they exhibit higher ER stress basal levels than normal cells. Nevertheless, different reports have demonstrated that under continuous stress conditions, cancer cells die because of excessive self-degradation during sustained stress and continuous progression of autophagy through CHOP-mediated apoptosis [171].

The downregulation of Beclin1 in different cancers could indicate that tumor development is closely related to Beclin1-induced autophagic cell death [178,184,185]. Beclin1 downregulation can significantly reduce autophagy to protect tumor cells from autophagic cell death, contributing to the continuous development of tumor cells [186]. If this is the case, induction of autophagy appears as a promising strategy, and drugs such as ABTL0812 or dihydroartemisinin that induce a robust and sustained ER stress could overpass the cytoprotective effect of UPR and induce autophagic cell death while being safer for nontumor cells which have lower basal stress levels and a broader margin to resist stress-induced cytotoxicity [172]. This hypothesis could explain the association of low Beclin1 expression with low survival and lymph node metastasis observed in CCA. Analyzing stress-marker expression could help identify those with higher basal levels of ER stress and potential targets for ER-stress-mediated cytotoxic autophagy induction. Hypoxia is an ER stress activator that uses autophagy as a protective mechanism and t has been proposed as a contributor factor for chemotherapeutic resistance of CCA cell lines [187], which would correlate with poor prognosis associated with HIF-1α, BNIP3 and PI3KC3 in CCA samples [173]. In these clinical settings, where ER stress markers are overexpressed in CCA cells, the induction of ER-stress-mediated cytotoxic autophagy could offer therapeutic benefits.

Analogous to autophagy inhibitors, multiple combinatory treatments including autophagy promoter drugs could offer a potentially successful strategy. ABTL0812 has already shown the potentiation of chemotherapy in lung [170] and endometrial cancer [169]. In mesothelioma [188] and multiple myeloma [189], ER-stress-mediated induction of cytotoxic autophagy induces the release of immunogenic signals that make tumors more immunogenic and targetable for the immune system. The induction of immunogenic cell death through ER-stress-mediated autophagy has been described for different drugs, including chemotherapy, being the basis for its combination with immunotherapies such as immune checkpoint inhibitors [190]. Pembrolizumab (anti-PD1) was tested in advanced biliary tract cancer with modest efficacy response rates [191]; therefore, combining ER-stress-mediated inductors of autophagy with chemotherapy to increase tumor immunogenicity and with anti-PD1 treatment could significantly increase the therapeutic ratio. Triple combination therapies are positioned as an effective anti-cancer strategy that already showed preclinical superiority controlling tumor growth in different cancer models of melanoma [192] and glioma [193]. A similar strategy is currently under clinical evaluation for metastatic pancreatic cancer patients, where a targeted therapy is combined with chemotherapy and pembrolizumab (NCT02826486), although it will be important to manage toxic effects that have been observed in previous trials ([194], NCT01767454), especially in those with advanced disease and worse health status.

Taking all data together, it could be proposed that during initial steps of oncogenic transformation, cholangiocytes inhibit autophagy to promote carcinogenesis and activate autophagy during CCA growth and dissemination. Nevertheless, in opposition with this hypothesis, some clinical data have shown lower autophagy marker expression associated with poor survival and lymph node metastasis, which reveals the complex relationship between autophagy and cancer cell fate. In human pancreatic cancer cells, Beclin1 genetic inhibition promotes autophagy and decreases gemcitabine-induced apoptosis [195]. This may indicate that autophagy could be regulated through Beclin1 interaction with pro- and anti-apoptotic proteins, without the necessity for the formation of the autophagic vesicle, highlighting the need to encourage researchers to uncover the molecular mechanisms regulating autophagy under oncogenic conditions, and precisely define whether autophagy is potentiated or suppressed in each specific case. Further research using preclinical models of CCA might shed some light on the role of autophagy in CCA and could help design novel therapeutic strategies based on patient stratification. Comparing ER stress and autophagy basal levels between CCA cells with different relevant mutations can be helpful for understanding how mutations regulate autophagy and in which subtype of patients autophagy inhibition would be more efficacious, such as KRAS and p53-mutated CCA, or in which autophagy activation would offer better outcome, such in those with impaired autophagy or higher expression of ER stress markers. Syngeneic murine models of CCA could also greatly help analyze autophagy status at different times of carcinogenic development, where the immune system also plays a relevant role. Moreover, preclinical in vivo studies will be necessary to test multiple combination treatments to be translated to clinics. Furthermore, the expression analysis of multiple autophagy markers could lead to better predict patient outcome and additionally identify which ones could benefit from inhibition or activation of autophagy.

Autophagy modulators in combination with chemotherapy, immunotherapy and targeted therapies are positioning as a promising strategy to increase therapeutic expectancy of cancer patients. Current treatment options for CCA are limited to chemotherapy, although with limited efficacy; thus, multiple combinations including autophagy modulators could offer a great opportunity to increase survival and quality of life of patients with the devastating disease.

## Figures and Tables

**Figure 1 cells-09-00614-f001:**
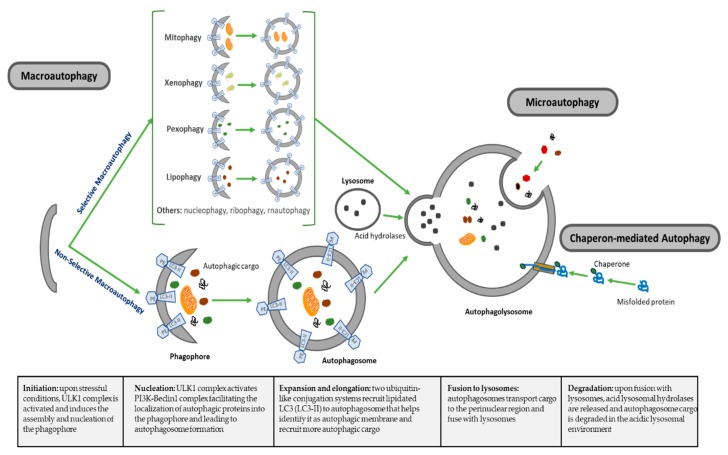
Schematic illustrating distinct types of autophagy. Intracellular components can enter the lysosome for degradation mainly by three autophagic pathways. 1) Macroautophagy: proteins, organelles and other cytosolic components are sequestered in a de novo-formed isolation membrane that expands and seals to form a double-membrane-bound vesicle, the autophagosome. Degradation occurs when autophagosomes fuse with lysosomes. Macroautophagy can be further subdivided into non-selective macroautophagy, where cytoplasmic components are engulfed into autophagosomes in a bulk manner; and selective macroautophagy, where distinct substrates (aggregate proteins, organelles or cellular components) are selectively targeted for degradation, giving rise to different types of selective autophagy depending on the autophagosomal cargo. 2) Microautophagy: invaginations at the surface of the lysosome or late endosomes trap cytosolic material, including proteins, and are then internalized after membrane scission and degraded in the lumen of the organelle. 3) Chaperone-mediated autophagy: soluble cytosolic proteins containing a targeting motif are recognized by the cytosolic chaperones which deliver the substrate to the membrane of the lysosome mediated by specific lysosomal-membrane bound receptors. The substrate protein unfolds and crosses the lysosomal membrane through a multimeric complex where it is degraded in the lysosomal lumen.

**Figure 2 cells-09-00614-f002:**
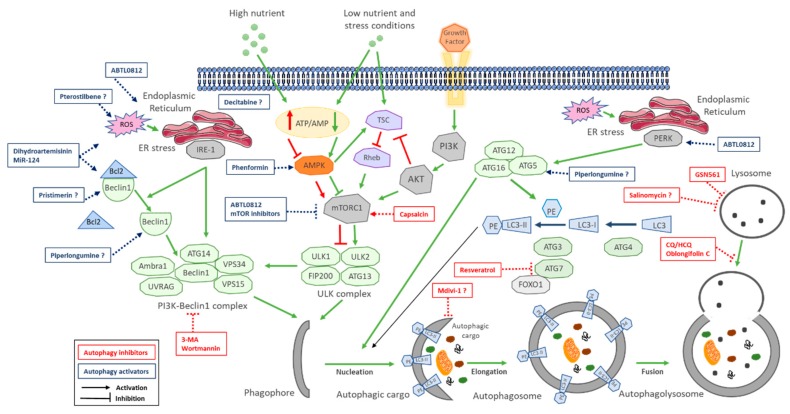
Schematic overview of the autophagy molecular pathway and target steps of its modulation. Upon nutrient or energy deprivation, AMP-activated protein kinase (AMPK) is activated, leading to mTORC1 inhibition and autophagy induction. Stress conditions activate the UPR (Unfolded Protein Response) mediated by PERK and IRE-1, which leads to the activation of autophagy. The ULK complex consists in ULK1, ULK2, FIP200 and ATG13. The PI3K-Beclin1 complex consists in VPS34, VPS15, Beclin1 and ATG14, or VPS34, Beclin1, UVRAG and Ambra1. These complexes mediate the generation of lipidated LC3 (LC3-II) and its incorporation into the phagophore membrane. The elongation of the phagophore ultimately closes and forms the autophagosome, which internalizes autophagosome cargo and fuses with lysosomes for cargo degradation and nutrient recycling. Current approaches to modulate autophagy in CCA target different steps. Autophagy inhibitors focus on inhibiting the last step, interfering with lysosome fusion or function, but other compounds target mTORC1 or other initiation steps. Autophagy activators act through targeting initial steps of autophagy, mTOR inhibition or ER-stress-induced autophagy.

**Table 1 cells-09-00614-t001:** Main types of autophagy.

Types of Autophagy	Features	Mechanism	Selectivity of Cargo
**Macroautophagy**	**Nonselective macroautophagy:** multistep process of nonselective degradation and recycling of cellular misfolded, aggregated or mutated proteins and damaged organelles. Mediated by the formation of autophagosomes and their fusion to lysosomes	Cytoplasm degraded in a bulk manner, including proteins, organelles and cytoplasmic components. Most-described autophagic process	Nonselective
	**Selective macroautophagy:** multistep process of selective degradation and recycling of specific targets organelles, proteins and cellular components. Mediated by the formation of autophagosomes and their fusion to lysosomes	Lipophagy: lipids droplets autophagic degradation	Selective
		Pexophagy: peroxisomes autophagic degradation	Selective
		Mitophagy: mitochondria autophagic degradation	Selective
		Xenophagy: microbes autophagic degradation	Selective
		Others: autophagic degradation of nucleus (nucleophagy), ribosomes (ribophagy), RNA (rnautophagy), etc.	Selective
**Microautophagy**	Direct uptake of cytoplasmic substances into the lysosomes for degradation. No autophagosome formation needed	Cytoplasmic substrates are engulfed via direct invagination, protrusion or septation of the lysosomal limiting membrane	Nonselective
**Chaperon-mediated autophagy (CMA)**	Uptake of soluble cytosolic proteins that are directly translocated across the lysosome membrane for degradation. No autophagosome formation needed	Chaperone-dependent recognition of specific sites in proteins to form the CMA substrate-chaperone complex, which is recognized by lysosomal membrane-bound receptors to unfold proteins and translocate them across lysosomal membranes	Highly selective for proteins

**Table 2 cells-09-00614-t002:** Preclinical studies with autophagy modulators in CCA. * Uncomplete mechanism of action.

**Autophagy Inhibitors**					
**Compound**	**Mechanism of Action**	**Preclinical Models**	**Effects on CCA**	**Level of Inhibition**	**Reference**
Wortmannin (cell permeable fungal metabolite) and 3-MA (synthetic 3 methyl adenine)	Specific class III PI3K (VPS34) inhibitors. VPS34 is needed to recruit Atg12-Atg5 conjugates to preautophagosomal structure	In vitro: QBC939, RBE and HCCC9810. In vivo: QBC939 xenografts	Apoptosis induction in vitro and inhibition of tumor growth, decreasing mRNA levels of ATG5 and Beclin1 in tumors	Initiation: inhibits Vps34 (class III PI3K) complex	Hou et al. 2011 [106]
Chloroquine (antimalaria agent)	Alters acidic environment of lysosomes, induces sustained ER stress and CHOP-mediated apoptosis	In vitro: CCKS1 and HuCCT1 cells	Attenuate invasive activity of CCA cells under starvation conditions and in TGF-β1-induced EMT	Fusion: Inhibits autophagosome fusion with lysosomes	Nitta et al. 2014 [126]
Capsaicin (major pungent component of chili peppers)	Interferes with NF-kB and AP-1 signaling	In vitro: QBC939, SK-ChA-1 and MZ-ChA-1. In vivo: QBC939 xenograft	Inhibition of 5-FU induced autophagy in vitro and in vivo via activation of PI3K/Akt/mTOR pathway, increasing sensitivity to 5-FU	Initiation: activates mTOR	Hong et al. 2015 [139]
Oblongifolin C (natural small molecule extracted from herbs)	Induces mitochondrial apoptotic pathway	In vitro: QBC939	Induces apoptosis and mitochondrial dysfunction	Fusion: Inhibits autophagosome fusion with lysosomes	Zang et al. 2016 [140]
Chloroquine (antimalaria agent)	Alters acidic environment of lysosomes, induces sustained ER stress and CHOP-mediated apoptosis	In vitro: QBC939 cells	Reduces antioxidant capacity of cells increases ROS and sensitizes cells to cisplatin	Fusion: Inhibits autophagosome fusion with lysosomes	Qu et al. 2017 [135]
Salinomycin (polyether antibiotic)	Interferes with WNT signaling and acts as potassium ionophore	In vitro: TFK-1 and EGI-1 cells. In vivo: s.c. and intrahepatic murine models KRAs and p53 mutated	Inhibits proliferation and transmembrane migration mediated by dysfunctional mitochondria in vitro and inhibits tumor growth in vivo	* Fusion: Inhibits autophagosome fusion with lysosomes	Klose et al. 2018 [138]
Chloroquine (antimalaria agent)	Alters acidic environment of lysosomes, induces sustained ER stress and CHOP-mediated apoptosis	In vitro: QBC939 cells	Induces apoptosis through activation of multiple death pathways and increases sensitivity to cisplatin	Fusion: Inhibits autophagosome fusion with lysosomes	Jia et al. 2018 [136]
Resveratrol (natural phenol, phytoalexin, produced by plants against infections)	Sirt1 agonist. Promotes deacetylation of FOXO1, blocking FOXO1 binding to Atg7	In vitro: QBC939 cells	Induces apoptosis by increasing oxidative stress and mitochondrial dysfunction.	Initiation: inhibits Foxo1-Atg7 activation	He et al. 2018 [30]
Mdivi1-selective Drp-1 inhibitor	Impedes mitochondrial dynamics	In vitro: KKU-156 and KKU-214	Potentiates cisplatin-induced apoptosis inducing mitochondrial dysfunction and ROS	* Elongation inhibits mitophagy	Tusskorn et al. 2019 [141]
GNS561 (lysosomotropic small molecule)	Lysosomal dysregulation through lysosome permeabilizes and releases hydrolytic enzymes to the cytosol	In vitro: HuCCT1 and RBE iCCAs. In vivo: chicken chorioallantoic membrane xenograft model	In vitro: reduces cell proliferation and induces apoptosis. In vivo: reduced tumor growth	Fusion: Inhibits lysosomal proteases	Brun et al. 2019 [137]
**Autophagy Activators**					
**Compound**	**Mechanism of Action**	**Preclinical Models**	**Effects on CCA**	**Level of Activation**	**Reference**
Decitabine (cytosine analog) DNA demethylating agent	DNA methyl transferase inhibitor	In vitro: TFK-1 and QBC939. In vivo: TFK-1 xenograft	Induces apoptosis and autophagy-dependent caspase-independent cell death in vitro and reduces tumor growth in vivo	* Initiation: epigenetic control of autophagy	Wang et al. 2014 [156]
Phenformin (biguanide compound paralog of metformin)		In vitro: RBE and Huh28. In vivo: RBE xenograft	Induces apoptosis and autophagy in vitro (Atg7, Atg5 and Beclin1 upregulation) and reduces tumor growth in vivo	Initiation: AMPK-mediated mTOR inhibition	Hu et al. 2017 [157]
Dihydroartemisinin (active compound from *Artemisia annua*)	ROS-mediated ER stress through DAPK activation promoting the disruption Beclin11-Bcl2	In vitro: KKU-452, KKU-023 and KKU-100, KKU-223 and MMNK-1	Induces apoptosis-dependent and autophagy-mediated apoptosis-independent cell death	Initiation: disruption of Beclin1-Bcl2	Thongchot et al. 2018 [154]
MiR-124 (associated with STAT3 regulation)	Targets EZH2 and STAT3 signaling pathway inducing ER stress	In vitro: HuCCT1, KMBC and MZChA1. In vivo MZChA1 transfected to stably express low levels of miR-124 or shEZH2	Induces autophagy-related cell death via EZH2-STAT3 signaling axis in vitro and tumor-suppressive function in vivo	Initiation: disruption of Beclin1-Bcl2	Ma et al. 2018 [29]
Piperlongumine (small molecule extracted from plants)	Inhibits the antioxidant enzyme glutathione S-transferase P, leading to elevated ROS via multiple pathways (p38/JNK, MAPK-C/EBO and NN-KB)	In vitro: HuCCT-1	Induces apoptosis and autophagy through ROS-activated Erk signaling	* Initiation: disruption of Beclin1-Bcl2	Chen et al. 2019 [123]
Pterostilbene (active constituent of blueberries; natural demethylated analogue of resveratrol	Involves overlap among intrinsic and extrinsic apoptotic pathway, cell cycle arrest, DNA damage, mitochondrial depolarization and autophagy	In vitro: RBE and HCCC-9810. In vivo: HCCC-9810	Induces dose-dependent and time-dependent cytotoxic effects and inhibits colony formation upregulating Beclin1, ATG5 and ATG7 and inhibits tumor growth in vivo	* Initiation: disruption of Beclin1-Bcl2	Wang et al. 2019 [152]
Pristimerin (triterpenoid isolated from herbs)	Has multiple targets (Li et al. 2018	In vitro: QBC and RBE. In vivo: QBC939 xenografts	Induces apoptosis and autophagy in dose-dependent manner, decreasing apoptosis-related proteins Bcl-2, Bcl-xL and porcaspase-3 in vitro and inhibits tumor growth in vivo	* Initiation: disruption of Beclin1-Bcl2	Sun et al. 2019 [153]
ABTL0812 (hydroxylated variant of linoleic acid)	Induces robust and sustained ER stress, and TRIB3-mediated Akt/mTOR axis inhibition, leading to cytotoxic autophagy	In vitro: EGI-1 and TFK-1	Induces ER stress-mediated cytotoxic autophagy (elevated ATF4, CHOP and TRIB3)	Initiation: mTOR inhibition and ER stress mediated autophagy initiation	Muñoz-Guardiola et al. 2020 [158]

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
