# Peer review of "Therapeutic Potential of Autophagy Modulation in Cholangiocarcinoma"

_cells, 2020, doi:10.3390/cells9030614_

Round 1

Reviewer 1 Report

The manuscript of Pérez-Montoyo is comprehensive, exhaustive, and very thorough review on mechanisms of autophagy and its significance in the pathogenesis and treatment of cholangiocarcinoma (CCA). The review is clear and well written and shed some new light on the mechanisms involving pathogenesis in CCA, and new suggestions and hints on possible use of autophagy as marker of disease progression or as novel putative target of therapy. The only suggestion from the reviewer is to add a table to summarize the differences on the different types of autophagy with their main features. Moreover, since the complex mechanisms of autophagy, a new figure showing the main characteristics of the different stages of autophagy could be added.    

Author Response

Dear reviewer,

I greatly appreciate your general comments regarding the efforts in collecting clinical and preclinical evidences involving autophagy molecular mechanisms and cholangiocarcinogenesis and the potential of autophagy regulation to develop novel, safer and more efficacious therapeutic strategies. 

Due to the relevance of autophagy in this review, I agree with you and believe that adding a table summarizing the differences on the different types of autophagy and their main features, along with a new figure showing the main characteristics of the different stages of autophagy, will improve the description of autophagy process and will facilitate the understanding of this biological process through the review.

The table below have been included in the manuscript in line 108 with track changes function of word activated.

TYPES OF AUTOPHAGY

FEATURES

MECHANISM

SELECTIVITY OF CARGO

Macroautophagy

Non-selective macroautophagy: multistep process of non-selective degradation and recycling of cellular misfolded, aggregated or mutated proteins and damaged organelles. Mediated by the formation of autophagosomes and their fusion to lysosomes

Cytoplasm degraded in a bulk manner, including proteins, organelles and cytoplasmic components. Most described autophagic process

Non-selective

Selective macroautophagy:  multistep process of selective degradation and recycling of specific targets organelles, proteins and cellular components. Mediated by the formation of autophagosomes and their fusion to lysosomes

Lipophagy: lipids droplets autophagic degradation.

Selective

Pexophagy: peroxisomes autophagic degradation

Selective

Mitophagy: mitochondria autophagic degradation

Selective

Xenophagy: microbes autophagic degradation

Selective

Others: autophagic degradation of nucleus (nucleophagy), ribosomes (ribophagy), RNA (rnautophagy) etc..

Selective

Microautophagy

Direct uptake of cytoplasmic substances into the lysosomes for degradation. No autophagosome formation needed

Cytoplasmic substrates are engulfed via direct invagination, protrusion, or septation of the lysosomal limiting membrane

Non-selective

Chaperon-mediated autophagy          (CMA)

Uptake of soluble cytosolic proteins that are directly translocated across the lysosome membrane for degradation.  No autophagosome formation needed

Chaperone-dependent recognition of specific sites in proteins to form the CMA substrate-chaperone complex, which is recognized by lysosomal membrane-bound receptors to unfold proteins and translocate them across lysosomal membranes

Highly selective for proteins

The new figure also illustrates the other types of autophagy that are summarized in the table and has been included in the manuscript in line 155 with track changes function of word activated.

Sincerely,  

Reviewer 2 Report

This manuscript clearly shows a big effort of the author in giving a very thorough overview of the field. This has a lot of value and can only be appreciated.

The author mostly sticks to summarizing the literature, rather than trying to synthesize a new perspective. Although a missed opportunity in my eyes, this is not always problematic. And despite thinking that this overview exersize is for the most part a valuable contribution to the literature, in this case, this approach leads to replicating (or amplifying) disruptive confusion within the literature. The key question concerns the dual role of autophagy in tumorigenesis and metastasis; whether there is rationale and/or clinical evidence for modulating it in this disease; and whether that modulation should be inhibition or activation. I think the author can do more with his knowledge to attempt to overcome this confusion. Or at least give a helpful context to better digest it. See lines 368-371: "assembling the pieces of this great puzzle..." Isn't that something the author should at least try to do here?

In one example (first half p8): Beclin-1 is a prognostic biomarker for iCC, meaning that high expression correlates to poor prognosis. Then, how do you square the correlation of low Beclin (mediating less autophagy) with reduced survival? As if that is not confusing enough, Ambra1 prognostication makes it even wilder. And in lines 617-625, Beclin-1 is a tumour suppressor, a biomarker for poor prognosis, a tumor promoter, and negatively correlates with metastasis, all in the same context (ref 27/28), without any helpful explanation or interpretation. More confusion, not less. [To my best understanding, a marker that is negative in normal tissue, expressed somewhat highly in early lesions, and lower in metastatic lesions (is this the case for Beclin-1?) will be very difficult to adopt in the clinic.]
At any rate, this needs a better explanation. What I want to know: a) Is all of this trustworthy science, and b) if so, what does it all mean?

Moreover, the discussion section reads just as a summary; and quite repetitive in a number of details.Critically, actual discussion — arguably one of the main reasons to read this review — is lacking. (The section seems to contribute just 2 extra references, for a very thin and gratuitous (cliche) invocation of checkpoint blockade immunotherapy.) 

In my opinion, at least the discussion needs to be re-written.

Other/smaller points:

* The claim that 'all' evidence and reports are included is highly unlikely, unnecessary, and should be reconsidered. 

* There are several repetitive passages, such as lines 73-78 with 52-59; 140-144 with 58-61

* Lines 249-251: this is the opposite of what I read in that reference (#83, O'Dell 2012 Cancer Res): "Together these results showed that autophagy is actively engaged in Kras-p53 IHCC cells, and that its inhibition using CQ attenuates cell proliferation, thus this approach may have value in the treatment of IHCC." This is very confusing.
Also, I would critically look at the entire paragraph, it is very long and could be clearer. 

Author Response

Dear reviewer,

I would like to greatly thank you for your comments on the manuscript. The criticism is fair and constructive, and your critical revision will lead to a much-improved manuscript, so I am pleased that you also provided me with the opportunity to address them through a resubmission.

The author mostly sticks to summarizing the literature, rather than trying to synthesize a new perspective. Although a missed opportunity in my eyes, this is not always problematic. And despite thinking that this overview exercise is for the most part a valuable contribution to the literature, in this case, this approach leads to replicating (or amplifying) disruptive confusion within the literature. The key question concerns the dual role of autophagy in tumorigenesis and metastasis; whether there is rationale and/or clinical evidence for modulating it in this disease; and whether that modulation should be inhibition or activation. I think the author can do more with his knowledge to attempt to overcome this confusion. Or at least give a helpful context to better digest it

I completely agree with the reviewer in that this manuscript does not propose new perspectives and does not fully address the specific roles of autophagy in cholangiocarcinogenesis. Clinical data is limited, and some of them show controversial results. Furthermore, pre-clinical evidences investigating underlying molecular mechanisms involving autophagy deregulation in CCA are limited to a few scientific reports, and most of pre-clinical studies testing the efficacy of autophagy modulators focus on validating the anti-cancer effect, rather than go deeper in the mechanism of action. Autophagy role in carcinogenesis is an intense area of research, and opposing data is still to be understood in all cancer types, even in those indications where research is in a more advance stage of development.

In agreement with the reviewer, a higher effort in trying to speculate with current data to present possible answers and propose future directions should have been made. Therefore, I have tried to further seek in literature in an attempt to propose a more detailed role of autophagy in CCA. Several reports pointed to an impaired autophagy during cholangiocytes oncogenic transformation, while increased autophagy markers are associated with poor prognosis, metastatic potential and less overall survival in CCA clinical samples, although some of them showing controversial results which I have tried to address, by looking at available clinical data on other indications such as pancreatic cancer or hepatocellular carcinoma.  I also found a couple of reports showing relevant data in CCA, further improving the discussion section. Finally, I tried to present possible therapeutic strategies that could be beneficious for CCA patients, proposing possible future directions hoping to encourage researchers towards this direction. 

Following reviewer´s suggestions, I have modified the manuscript addressing your specific questions and I have re-written  the discussion section trying to address main questions such as dual role of autophagy in CCA, whether its modulation should be inhibition or activation as well as  clinical and pre-clinical evidences reinforcing the therapeutic potential of autophagy modulation 

In one example (first half p8): Beclin-1 is a prognostic biomarker for iCC, meaning that high expression correlates to poor prognosis. Then, how do you square the correlation of low Beclin (mediating less autophagy) with reduced survival? As if that is not confusing enough, Ambra1 prognostication makes it even wilder. And in lines 617-625, Beclin-1 is a tumour suppressor, a biomarker for poor prognosis, a tumor promoter, and negatively correlates with metastasis, all in the same context (ref 27/28), without any helpful explanation or interpretation. More confusion, not less. [To my best understanding, a marker that is negative in normal tissue, expressed somewhat highly in early lesions, and lower in metastatic lesions (is this the case for Beclin-1?) will be very difficult to adopt in the clinic.]

At any rate, this needs a better explanation. What I want to know: a) Is all of this trustworthy science, and b) if so, what does it all mean?

I completely agree with the reviewer that current data on Beclin1 expression for iCCA is limited, confusing and far for being clinically useful. Moreover, the way I presented Beclin1 data, along with other positive regulators of autophagy such as Ambra1 was confusing and not well structured. In an effort to significantly improve the understanding of the role of Beclin1 both, mechanistically and therapeutically, I have modified text in line 393 (with track changes function of word activated), and re-wrote de discussion trying to address opposing data on Beclin1 expression, as well as propose improved methods for assessing autophagy in clinical samples and their correlation with prognosis, metastasis and survival

The new paragraph explaining Beclin1 data in “4. AUTOPHAGY MODULATION IN CHOLANGIOCARCINOMA” section is now as follows:

“Beclin1 plays a relevant role in linking autophagy, apoptosis and differentiation and its inactivation and consequent deficiency in autophagy was correlated with malignant transformation, although existing data on the prognostic role of Beclin1 in human carcinomas is contradictory, appearing under and overexpressed in distinct human carcinomas [49,127,128]. Moreover, sSeveral studies have shown the significance of Beclin1 in iCCA [27,28] and eCCA [28], revealing its potential prognostic value for CCA. Beclin1 was found markedly expressed in iCCA samples compared with normal bile duct epithelium [27], and among Beclin1 positive samples, those with and low Beclin1 expression was were significantly associated with lymph node metastasis, worse overall survival and less disease-free survival [27,28]. Moreover, in a lymph node negative CCA subgroup, Beclin1 was higher than in the lymph node positive subset, suggesting that Beclin1 inactivation and therefore impaired autophagy, might promote malignant phenotypes. Interestingly, a stratified survival analysis in patients with Beclin1 low expression, iCCA patients showed a worse overall survival and progression-free survival than eCCA [28], which may indicate a higher implication of autophagy in iCCA subgroup of patients. Nevertheless, low Beclin1 levels show correlation with poor prognosis in both subtypes [28]. This clinical data is in contradiction with other reports that indicate an exacerbated autophagy in CCA samples and its association with lower survival and tumor dissemination. Ambra1, a positive regulator of Beclin1 dependent program of autophagy, positively correlated with SNAIL expression in CCA patients. SNAIL is a hallmark of EMT activation which is in accordance with the in vitro increased invasive potential mediated by autophagy in TGF-β1/SNAIL induced EMT [126]. These opposing results underscore the need to clearly define the type of studies that would help to discern whether the presence of autophagy related markers are associated with impaired or increased autophagic flux and additional expression studies of other markers such as LC3-II, p62, PI3Ks or ATGs could add significant value.”

I understand reviewer´s concern about the origin of the scientific reports, since I have come across several untrustworthy publications myself, but I have tried to be careful in the selection of the papers based on authors and their affiliation, scientific journals where papers were published and their citation in other publications.  

Moreover, the discussion section reads just as a summary; and quite repetitive in a number of details.Critically, actual discussion — arguably one of the main reasons to read this review — is lacking. (The section seems to contribute just 2 extra references, for a very thin and gratuitous (cliche) invocation of checkpoint blockade immunotherapy.)

In my opinion, at least the discussion needs to be re-written.

I agree with the reviewer in that the discussion can be significantly improved and try to make a bigger effort to emphasize hints that could help explain the relevance of autophagy in CCA. Below you can find the new discussion in which I have tried to put all puzzle pieces together and tried to shed light on the current data to open novel research strategies that could help better understand this process, improve patient selection and therapeutic options for this patients. 

I further sought in literature and provided the discussion with additional bibliography that help better understand the whole message and guide possible future research.

I have included the new discussion section in line 700 (with track changes function of word activated), which now is as follows:

“ Autophagy is a tightly orchestrated multi-step catabolic process generally considered a pro-survival mechanism, which allows cells to recover homeostasis under stressful conditions by controlling energy and nutrient balance [17,18]. The presence of multiple checkpoints within the autophagic process increases the possibilities of disturbing autophagy and develop different human diseases including cancer, although it also offers multiple target points for therapeutic approaches [19]. The precise molecular mechanisms linking autophagy and cancer cell fate are still to be determined, although numerous reports addressed to uncover these molecular mechanisms have been released during the last decades. Autophagy may act as tumor suppressor at the early stages of cancer development, impeding the appearance of oncogenic mutations through the clearance of impaired macromolecules and organelles that cause DNA damage and chromatin instability [43,44]. When the autophagic process is impaired, the accumulation of p62 aggregates, defective mitochondria, poorly folded proteins and increased intracellular ROS promote malignant transformation [43,51,52].

In CCA, several evidences strongly suggest a deregulated autophagy at the initial steps of cholangiocarcinogenesis, where defective autophagy would allow oncogenic transformation. Supporting this theory, Greer et al showed a genetic risk of CCA linked to ATG7 deficiency and therefore autophagy impairment, mediated by a lack of lipidation activity and p62 accumulation compared with wild type ATG7 carriers [33]. Moreover, precursor BilIN lesions showed higher levels of LC3-II and p62 compared with normal biliary ducts [32] indicative of uncomplete autophagic process, reinforcing the theory of autophagy inhibition as contributor of carcinogenic transformation. Several genetic alterations commonly observed in CCA have also been linked to autophagy inhibition in other types of cancers, such as c-Met [95], FGFR gain of function [100,101] or HDAC6 overexpression [111]. These genetic alterations could mediate cholangiocyte oncogenic transformation through the inhibition of autophagy, cooperating with their proliferative and pro-survival derived effects. It has been demonstrated that continuous IL-6 secretion mediated by STAT3 inhibits autophagy, contributing to arsenic carcinogenesis in lung cells during carcinogenesis [80] strengthening the idea of impaired autophagy during CCA establishment that could also be taken place in the inflammatory subtype of CCA.

Autophagy can also act promoting tumor growth on stablished tumors serving as an adaptive and pro-survival mechanism against the extreme tumor microenvironment conditions such as lack of oxygen, limited nutrients and high metabolic rate [53-55]. Thongchot and colleagues found a positive correlation between HIF-1α (Hypoxia-inducible factor 1-α) with BNIP3 (pro-apoptotic member of Bcl2 family) and PI3KC3 (component of Beclin1-PI3K complex), that associated with poor prognosis and lymph node metastasis in CCA samples [179], which reflects an hypoxic stress that activates autophagy as pro-survival and invasive mechanism. Similarly, RAS-mutated cells have been defined as addicted to autophagy by maintaining oxidative metabolism and glycolysis, underpinning growth, survival, invasion and metastasis [58,59]. RAS appears frequently mutated in CCA [72], suggesting these cells could also have high dependence on autophagy for survival. Supporting this idea, a transgenic murine model of iCCA carrying KRAS and p53 genetic alterations, which recapitulates histopathologic features of human disease, showed actively engaged autophagy [83]. Treatment of primary cells derived from intrahepatic murine tumors with CQ led to LC3-II accumulation and induced cancer cell death, revealing an active autophagy in these tumors. Interestingly, when autophagy is impaired in these cells by ATG7 deletion, mice died for inflammation rather than for tumor-derived effects such as lung or liver metastatic, further reinforcing the idea of autophagy activation as mediator of survival and growth in CCA [83].

The use of autophagy inhibitors such as HCQ or CQ arises a very promising strategy to treat different cancers, especially those with autophagy dependence for growing and dissemination. In KRAS-driven cancers, autophagy-dependent production of secreted factors facilitates invasion [59], where EMT has a prominent role. EMT induced by TGF-β in CCA cells was shown to mediate a higher invasive capacity [125], and inhibition of autophagy impaired invasiveness in vitro mediated by EMT induction, which highlights the importance of autophagy for increasing CCA metastatic potential. Moreover, this in vitro data correlates with higher expression of autophagy related markers in CCA patients with lymph node metastasis such as Ambra1 [126], which also correlates with SNAIL expression, a master regulator of EMT. Another advantage of inhibiting autophagy relies on the blockage of the protective mechanism mediated by autophagy activation induced by different drugs. A wide variety of anticancer compounds induce protective autophagy in CCA [145-147] including chemotherapy [106,141] and the inhibition of autophagy accelerated apoptosis and chemosensitized CCA cells. This opens different possibilities to design combinatory treatments that could block this protective autophagy and enhance the therapeutic effects of different drugs, in addition to diminish tumor dissemination. This is the rationale behind a clinical trial currently ongoing for CCA patients, in which HCQ is administered in combination with a selective SK inhibitor (ABC294640), previously shown to induce protective autophagy in cancer [131]. Inhibiting autophagy would block the activation of autophagy as a mechanism of resistance, and could potentially decrease CCA metastatic potential, therefore clinical results of this study would be of a great help for further design of novel therapeutic strategies involving autophagy inhibitors in CCA.

Beclin1 has been defined as tumor suppressor and is a critical factor in autophagy initiation, directly interacting with pro-survival and pro-death factors, thus being involved in cell fate decision making [44,49,174]. In CCA, different reports analyzing the potential role of Beclin1 as prognostic marker have been released, although showing some contradictory results. Beclin1 was found overexpressed in CCA samples compared with normal biliary duct cells, and withing Beclin1 positive CCA samples, low Beclin1 was associated with poor prognosis and lymph node metastasis [27,28]. Interestingly, low Beclin1 expression was associated with poor prognosis and less overall survival in both, iCCA and eCCA patients, although iCCA had an inferior overall survival compared with eCCA patients. In opposition to this data, Ambra1, a positive regulator of Beclin1, showed higher expression in CCA patients with lymph node metastasis and poor survival [126].

Similar to CCA, Beclin1 expression in different cancers is differently associated with prognosis, metastasis and survival [49, 127,128]. In ovarian carcinomas, decreased expression of Beclin1 was correlated with histological grade, advanced clinical stage and shortened patient survival and inversely correlated with Bcl-xL expression, showing the low Beclin1/high Bcl-xL group the lowest survival rate [180]. In breast carcinomas, low expression of Beclin1 may contribute to the development and progression of breast cancer [49]. Conversely, high beclin1 expression was found predictive of poor prognosis in nasopharyngeal carcinoma [181] and Beclin1 and LC3 high expression correlated with tumor stage, metastasis and survival in pancreatic [182] and colorectal [183] cancers. Recent studies addressing the potential of Beclin1 expression as prognostic factor in different cancers have emphasized the necessity to combine Beclin1 expression with other autophagy related proteins such as HIF-1α (Hypoxia Inducible Gene 1 α), Bcl2 family proteins Bcl-xL and BNIP3, PI3KC3 or ATGs, to increase its clinical value.

In recent studies, the low Beclin1/ Bcl-xL high population, but not the low Beclin1/ Bcl-xL low population of HCC patients was associated with most aggressive disease and tumor differentiation [184], and similar results were observed between Beclin1 and apoptotic markers Bcl-2 and Bax [185] and between Beclin1 and HIF-1α [186]. In a histopathological retrospective study on iCCA clinical samples, ARID1A, CA9 and IDH1 were found highly expressed in iCCA tumor tissues, but only high Beclin-1/high ARID1A population were strongly associated with poor prognosis, lower survival rate and a worse recurrence rate than patients with low Beclin-1/low ARID1A expression [187]. This recently published study seems to be in contradiction with previously published reports where low Beclin1 associated with poor prognosis [27,28]. All these data underline a need for clearly defined specific marker combinations that could predict CCA prognosis, metastasis and survival, and that could potentially serve to stratify patients for specific combinatory treatments involving autophagy modulators. For example, expression analysis of p62 and LC3-II protein levels could significantly help to identify whether autophagy is engaged or impaired. Higher Beclin1 levels could indicate increased autophagic activity, but if is accompanied with p62 accumulation would indicate autophagy impairment probably due defects in last steps in autophagosome degradation. A good example of the usefulness of the detection of multiple markers in CCA was the positive correlation found between HIF-1α with BNIP3 and PIK3CA, indicative of high autophagic activity and related with poor prognosis [179], therefore positioning this population of patients as potential targets for autophagy inhibition therapeutics.

The induction of autophagy as therapeutic approach to treat CCA is also showing promising results. The induction of ER stress mediated cytotoxic autophagy by increasing intracellular dihydroceramides (Dh-Cer) content has been proposed as a safe and efficient way to induce autophagy mediated apoptosis in cancer cells. Resveratrol [144], which in CCA acts inhibiting autophagy, and THC [174] induce an increase in Dh-Cer in cancer cells by inhibiting dihydroceramide desaturase (Des-1), which is responsible for ER stress mediated autophagy promotion. Similarly, ABTL0812 induces impairment of Des-1 activity, resulting in the accumulation of Dh-Cer and activation of UPR response, which in combination with TRIB3-mediated AKT/mTOR axis inhibition, triggers cytotoxic autophagy in CCA cells [158,160]. Interestingly, Des1 expression was found to be upregulated in CCA cell lines compared with their non-tumor counterparts NHC3 cells [158], correlating with previous reports [188], where Des1 was found overexpressed in CCA tissue compared with normal biliary tract tissue. Cancer cells have evolved to use the UPR to survive the ER stress induced by the hostile conditions of tumor microenvironment (hypoxia, low glucose, intracellular acidification, etc.), therefore they exhibit higher ER stress basal levels than normal cells. Nevertheless, different reports have demonstrated that under continuous stress conditions, cancer cells die because of excessive self-degradation during sustained stress and continuous progression of autophagy through CHOP-mediated apoptosis [171].

The downregulation of Beclin1 in different cancers could indicate that tumor development is closely related to Beclin1-induced autophagic cell death [183,188]. Beclin1 downregulation can significantly reduce autophagy to protect tumor cells from autophagic cell death, contributing to the continuous development of tumor cells [190]. If this is the case, induction of autophagy appears as promising strategy and drugs such as ABTL0812 or dihydroartemisinin that induce a robust and sustained ER stress could overpass the cytoprotective effect of UPR and induce autophagic cell death, while being safer for not tumor cells which have lower basal stress levels and a broader margin to resist stress-induced cytotoxicity [172]. This hypothesis could explain the association of low Beclin1 expression with low survival and lymph node metastasis observed in CCA. Analyzing stress marker expression could help identify those with higher basal levels of ER stress and potential targets for ER stress mediated cytotoxic autophagy induction. Hypoxia is an ER stress activator that uses autophagy as a protective mechanism that has been proposed as a contributor factor for chemotherapeutic resistance of CCA cell lines [191], which would correlate with poor prognosis associated to  HIF-1α, BNIP3 and PI3KC3 in CCA samples [179]. In these clinical settings, where ER stress markers are overexpressed in CCA cells, the induction of ER stress mediated cytotoxic autophagy could offer therapeutic benefits.

Analogous to autophagy inhibitors, multiple combinatory treatments including autophagy promoter drugs could offer as a potentially successful strategy. ABTL0812 has already shown potentiation of chemotherapy in lung [170] and endometrial cancer [169]. In mesothelioma [175] and multiple myeloma [176], ER stress mediated induction of cytotoxic autophagy induces the release of immunogenic signals that make tumors more immunogenic and targetable for immune system. The induction of immunogenic cell death through ER stress mediated autophagy has been described for different drugs, including chemotherapy, being the basis for its combination with immunotherapies such as immune checkpoint inhibitors [177]. Pembrolizumab (anti-PD1) was tested in advanced biliary tract cancer with modest efficacy response rates [178], therefore combining ER stress mediated inductors of autophagy with chemotherapy to increase tumor immunogenicity and with anti-PD1 treatment could significantly increase the therapeutic ratio. Triple combination therapies are positioning as an effective anti-cancer strategy that already showed preclinical superiority controlling tumor growth in different cancer models of melanoma [192] and glioma [193]. Similar strategy is currently under clinical evaluation for metastatic pancreatic cancer patients, where a targeted therapy is combined with chemotherapy and pembrolizumab (NCT02826486), although it will be important to manage toxic effects that have been observed in previous trials ([194], NCT01767454), especially in those with advanced disease and worse health status.

Taken all data together, it could be proposed that during initial steps of oncogenic transformation, cholangiocytes inhibit autophagy to promote carcinogenesis and activate autophagy during CCA growth and dissemination. Nevertheless, in opposition with this hypothesis, some clinical data have shown lower autophagy marker expression associated with poor survival and lymph node metastasis, which reveals the complex relationship between autophagy and cancer cell fate. In human pancreatic cancer cells, Beclin1 genetic inhibition promotes autophagy and decreases gemcitabine–induced apoptosis [195]. This may indicate that autophagy could be regulated through Beclin1 interaction with pro and anti-apoptotic proteins, without the necessity for the formation of the autophagic vesicle, highlighting the need to encourage researchers to uncover the molecular mechanisms regulating autophagy under oncogenic conditions, and precisely define whether autophagy is potentiated or suppressed in each specific case. Further research using preclinical models of CCA might shed some light on the role of autophagy in CCA and could help design novels therapeutic strategies based on patient stratification. Comparing ER stress and autophagy basal levels between CCA cells with different relevant mutations or by generating relevant mutants can be helpful for understanding how mutations regulate autophagy and in which subtype of patients autophagy inhibition would be more efficacious, such as KRAS and p53 mutated CCA, or in which autophagy activation would offer better outcome, such in those with impaired autophagy or higher expression of ER stress markers. Syngeneic murine models of CCA could also greatly help analyze autophagy status at different times of carcinogenic development, where the immune system also plays a relevant role. Moreover, preclinical in vivo studies will be necessary to test multiple combination treatments to be translated to clinics. Furthermore, the expression analysis of multiple autophagy markers could lead to better predict patient outcome and additionally identify which ones could benefit from inhibition or activation of autophagy. 

Autophagy modulators in combination with chemotherapy, immunotherapy and targeted therapies is positioning as a promising strategy to increase therapeutic expectancy of cancer patients. Current treatment options for CCA are limited to chemotherapy although with limited efficacy, thus multiple combinations including autophagy modulators could offer a great opportunity to increase survival and quality of life of patients with devastating disease.”

 The claim that 'all' evidence and reports are included is highly unlikely, unnecessary, and should be reconsidered.

The reviewer is completely right. Affirming that “all” evidences are included is very unnecessary and should has not been included. I have removed this affirmation from the manuscript, and simply affirm that collects clinical and preclinical evidences, stating as follows:

“ This review collects clinical and preclinical scientific reports involving autophagy modulation in CCA, putting all puzzle pieces together to try to shed light on the current knowledge of this therapeutic strategy for treating this aggressive disease.”

* There are several repetitive passages, such as lines 73-78 with 52-59; 140-144 with 58-61

I agree with the reviewer and have simplified the text in line 76 (with track changes function of word activated) as follows:

 “Macroautophagy (referred hereafter as autophagy), is a highly conserved catabolic process for recycling elderly, toxic or damaged intracellular components, mediated by the formation of autophagosomes that ultimately fuse to lysosomes for degradation”

I also simplified the text in line (with track changes function of word activated) as follows:

 “Although autophagy was initially defined as a pro-survival cellular mechanism due to its role in maintaining homeostasis under stressful conditions, several reports have revealed its dual role in cancer, and the therapeutic potential of its modulation”

* Lines 249-251: this is the opposite of what I read in that reference (#83, O'Dell 2012 Cancer Res): "Together these results showed that autophagy is actively engaged in Kras-p53 IHCC cells, and that its inhibition using CQ attenuates cell proliferation, thus this approach may have value in the treatment of IHCC." This is very confusing.

Also, I would critically look at the entire paragraph, it is very long and could be clearer.

I would like to apologize for this mistake, since the affirmation is wrong and not real, and following the reviewer´s suggestions, I have re-written the entire paragraph trying to address previous confusing sentence.

I have re-written the entire paragraph in line 292 (with track changes function of word activated) as follows:

“To date, different genes have been related to cholangiocarcinogenesis. Activating KRAS mutations can be found in up to 40% of CCAs, with major prevalence in dCCA and associated to a worse patients' prognosis [72]. In a small study on 54 clinical samples of iCCA, 7.4% of cases were KRAS mutated and associated with higher tumor stage and worse long-term overall survival, as well as a greater likelihood of lymph node involvement [84]. Moreover, in a murine model of iCCA development harboring KRAS mutation and p53 inactivation, two of the most common genetic alterations in CCA [72, 90], KRAS mutation collaborates with p53 deletion to cause hepatic transformation and reduced survival [83]. This murine model recapitulates histopathologic features of human iCCA and shows high basal levels of autophagy associated with tumor growth. Inhibition of autophagy with chloroquine (CQ) inhibited the growth of these cells and accumulated LC3-II, indicative of an active autophagy directly involved in tumor progression [83]. This data correlates with human iCCA cell lines mutated in KRAS and with p53 deficiency, which show elevated autophagy compared with normal iCCA cells, and CQ also inhibited the growth of these cells [91], similar situation described for pancreatic and lung cancers [56,86–89]. No specific RAS inhibitors have been developed so far and targeted therapies aiming to modulate KRAS downstream pathways such as MEK1/2 inhibitor selumetinib are in development for CCA, pointing to the potential combination with autophagy inhibitors to improve their therapeutic potential [4,92]."

I hope that through this letter, clearer explanations and modifications of the manuscript, I will have addressed all major and minor concerns of the reviewer as best as possible and to his/her satisfaction.

Sincerely
